# GEOMETRICALLY REGULARIZED AUTOENCODERS FOR NON-EUCLIDEAN DATA

**Cheongjae Jang[1], Yonghyeon Lee[2,3], Yung-Kyun Noh[1,3], Frank Chongwoo Park[2,4]**
[1]Hanyang University, [2]Seoul National University, [3]Korea Institute for Advanced Study,
[4]Saige Research
`cjjang@hanyang.ac.kr`, `ylee@kias.re.kr`, `nohyung@hanyang.ac.kr`,
`fcp@snu.ac.kr`

## ABSTRACT

Regularization is almost *de rigueur* when designing autoencoders that are sparse and robust to noise. Given the recent surge of interest in machine learning problems involving non-Euclidean data, in this paper we address the regularization of autoencoders on curved spaces. We show that by ignoring the underlying geometry of the data and applying standard vector space regularization techniques, autoencoder performance can be severely degraded, or worse, training can fail to converge. Assuming that both the data space and latent space can be modeled as Riemannian manifolds, we show how to construct regularization terms in a coordinate-invariant way, and develop geometric generalizations of the denoising autoencoder and reconstruction contractive autoencoder such that the essential properties that enable the estimation of the derivative of the log-probability density are preserved. Drawing upon various non-Euclidean data sets, we show that our geometric autoencoder regularization techniques can have important performance advantages over vector-spaced methods while avoiding other breakdowns that can result from failing to account for the underlying geometry.

## 1 INTRODUCTION

Regularization is almost *de rigueur* when designing autoencoders that are sparse and robust to noise. With appropriate regularization, autoencoders enable representations useful for downstream applications (Bengio et al., 2013), generate plausible data samples (Kingma & Welling, 2013; Rezende et al., 2014), or even obtain information on the data-generating probability density (Vincent et al., 2010; Rifai et al., 2011b).

Existing work on autoencoder regularization has mostly been confined to vector spaces, i.e., the data are assumed to be drawn from a vector space. On the other hand, a significant and growing number of problems in machine learning involve data that is non-Euclidean (in some past cases the fact that the data was non-Euclidean was not recognized or ignored). Bronstein et al. (2017) reviews several deep neural network architectures and modeling principles to explicitly deal with data defined on non-Euclidean domains, e.g., data collected from sensor networks, social networks in computational social sciences, or two-dimensional meshes embedded in the three-dimensional space. Other works have also addressed manifold-valued data including human mass and shape data (Kendall, 1984; Freifeld & Black, 2012), directional data (Mardia, 2014), point cloud data (Lee et al., 2022), and MRI imaging data (Fletcher & Joshi, 2007; Banerjee et al., 2015), with several deep neural networks proposed to handle such data in a coordinate-invariant way (Huang & Van Gool, 2017; Chakraborty et al., 2020).

The fundamental idea behind these works is that the geometrical structure of the curved space from which the non-Euclidean data are drawn needs to be accounted for properly, so that the output of any deep learning network applied to such input data should not depend on the particular choice of coordinates used to parametrize the data. Ignoring the underlying geometry of the data and simply applying standard vector space techniques can severely degrade performance, or worse, cause training to fail. Autoencoder training and its regularization are no exception.

For example, consider autoencoder training on a set of data points on a sphere as shown in Figure 1. When using spherical coordinate representations as inputs to train an autoencoder with a contractive regularization, the trained reconstruction function can heavily depend on the choice of coordinates. Moreover, it often fails to learn the correct contractive directions toward data-dense regions, especially near the singularity (or the spherical coordinate origins). On the other hand, an autoencoder that properly reflects the spherical constraints can recover those directions successfully and show results almost invariant to the choice of coordinates.

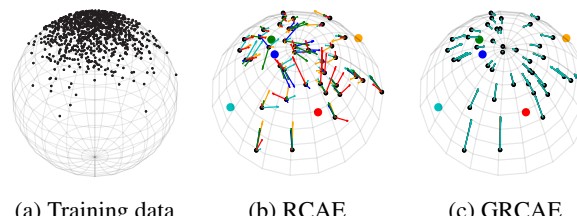

(a) Training data  (b) RCAE  (c) GRCAE

Figure 1: Autoencoder training on spherical data sampled from the von Mises-Fisher (vMF) distribution. We train the reconstruction contractive autoencoder (RCAE) and the geometric RCAE (GRCAE). For (b)-(c), we plot the reconstruction directions of the autoencoders trained using representations obtained from different coordinate choices. The results from each coordinate choice are color-coded along with corresponding spherical coordinate origins. (See Appendix E for more details.)

In this paper we address the regularization of autoencoders on curved spaces. Any loss function used to train or regularize the autoencoder should be formulated in a coordinate-invariant way, i.e., invariant to the choice of local coordinates used to parametrize the data, and instead depend only on the intrinsic properties of the curved space such as curvature or the choice of metric.

Assuming that both the data space and latent space can be modeled as Riemannian manifolds, we show how to construct regularization terms and objective functions in a coordinate-invariant way. We also develop geometric generalizations of the denoising autoencoder (DAE) and reconstruction contractive autoencoder (RCAE) such that the essential properties that enable the estimation of the score, i.e., the log-derivative of the data-generating density, are preserved. We provide some applications that use this property, such as sampling, clustering, and filtering for non-Euclidean data, and also show that the proposed autoencoders can obtain useful representations for non-Euclidean data, especially when noise exists in data. Drawing upon various non-Euclidean data sets, we show that our geometric autoencoder regularization techniques can have important performance advantages over vector-spaced methods – in some cases by significant margins – while avoiding other breakdowns that can result from failing to account for the underlying geometry.

The paper is organized as follows. We describe regularized autoencoders for Euclidean data in Section 2 and propose their coordinate-invariant generalizations to non-Euclidean data in Section 3. We then provide autoencoder training case studies using non-Euclidean data sets in Section 4.

## 2 REGULARIZED AUTOENCODERS FOR EUCLIDEAN DATA

Mathematically, an autoencoder can be represented as the composition of two mappings $f : \mathbb{R}^D \to \mathbb{R}^d$ (the encoder) and $g : \mathbb{R}^d \to \mathbb{R}^D$ (the decoder), i.e., $r = g \circ f : \mathbb{R}^D \to \mathbb{R}^D$ with the space of hidden variables $\mathbb{R}^d$. Assume there exists a data-generating probability density $\rho : \mathbb{R}^D \to \mathbb{R}$ from which data points on $\mathbb{R}^D$ are drawn. Autoencoder training in a vector space can then be formulated as minimizing the reconstruction error $\int_{\mathbb{R}^D} \|x - g(f(x; \theta_1); \theta_2)\|^2 \rho(x) \, dx$ over $\theta = (\theta_1, \theta_2)$, where $x \in \mathbb{R}^D$ denotes an input variable, $\theta_1$ and $\theta_2$ are respectively the parameter sets of the maps f and g, and $\|\cdot\|^2$ is the squared Euclidean norm. In the typical case where $d < D$, the autoencoder engages in a type of dimensionality reduction. By disregarding the assumption that $d < D$, autoencoders have been modeled in the form of deep artificial neural networks accompanied by certain regularization terms to learn useful representations of the data (Bengio et al., 2007; Vincent et al., 2008; 2010; Ranzato et al., 2007; 2008; Kingma & Welling, 2013; Rezende et al., 2014; Rifai et al., 2011b;a). For a more comprehensive review of autoencoders, we refer the reader to Goodfellow et al. (2016).

In the meantime, the effects of regularization have been investigated in some detail in Alain & Bengio (2014) for the denoising autoencoder (DAE) (Vincent et al., 2010) and the reconstruction contractive autoencoder (RCAE). They point out that these regularization methods reduce the autoencoder's sensitivity to the input, while the reconstruction error increases the autoencoder's sensitivity to variations along the region of the highest density in the data space. Reconstruction and regularization together successfully capture variations in such regions while ignoring variations that

are orthogonal to those and obtain information on the data-generating probability density. Our focus will be on these two types of regularized autoencoders, i.e., the DAE and the RCAE, but the methods described in this paper are easily generalizable to other autoencoders.

In the standard vector space formulation of the DAE, an input $x \in \mathbb{R}^D$ is assumed to have been corrupted by some noise density $q(\tilde{x}|x)$ to $\tilde{x} \in \mathbb{R}^D$, i.e., $\tilde{x} \sim q(\tilde{x}|x)$. We then seek the reconstruction function $r = \mathrm{g} \circ \mathrm{f} : \mathbb{R}^D \to \mathbb{R}^D$ that minimizes

$$\min_r \int_{\mathbb{R}^D} E_{q(\tilde{x}|x)} \left[ \|r(\tilde{x}) - x\|^2 \right] \rho(x) \, dx, \tag{1}$$

where $E_{q(\tilde{x}|x)}\left[ \, \cdot \, \right]$ denotes the expectation with respect to the noise density $q(\tilde{x}|x)$. A trivial identity mapping $r(x) = x$ can be avoided due to the injected noise.

For the vector space formulation of the RCAE, the objective function is

$$\min_r \int_{\mathbb{R}^D} \left( \|r(x) - x\|^2 + \sigma^2 \mathrm{Tr} \left( \left( \frac{\partial r}{\partial x} \right)^\top \left( \frac{\partial r}{\partial x} \right) \right) \right) \rho(x) \, dx, \tag{2}$$

where $\sigma^2$ is a scalar weighting coefficient. The second term in (2) acts as a regularization term and can be interpreted as the Dirichlet energy of $r : \mathbb{R}^D \to \mathbb{R}^D$, measuring how variable the mapping $r$ is (Belkin & Niyogi, 2003; Solomon et al., 2013). Minimizing this term induces the contraction of the mapping $r$ (e.g., in the absence of the reconstruction error term in (2), an extreme contraction of $r$ = constant would be obtained), preventing $r(x)$ from becoming the identity mapping $r(x) = x$. Note that replacing $\left( \frac{\partial r}{\partial x} \right)$ with $\left( \frac{\partial \mathrm{f}}{\partial x} \right)$ in (2) reduces to the objective function for the contractive autoencoder (CAE) (Rifai et al., 2011b).

For both the DAE under a Gaussian corruption process $\tilde{x} \sim N(x, \sigma^2 I)$ and the RCAE with $\sigma$ small, the derivative of the log-probability density, which is also referred to as the score, can be estimated from the optimized $r(x)$ as follows (Alain & Bengio, 2014):

$$\frac{\partial \log \rho(x)}{\partial x} = \frac{1}{\rho} \frac{\partial \rho}{\partial x}(x) = \frac{r(x) - x}{\sigma^2} + \mathcal{O}(\sigma^2). \tag{3}$$

## 3    REGULARIZED AUTOENCODERS FOR NON-EUCLIDEAN DATA

In this section, we address the problem of autoencoder training for the case where the data points are drawn from an *a priori* known non-Euclidean space $\mathcal{M}$, possibly with another non-Euclidean latent space $\mathcal{N}$. We formulate the reconstruction error and regularization terms in a coordinate-invariant way using notions from Riemannian geometry. (We provide some mathematical backgrounds required for our formulations in Appendix A.) We then show that, as in the Euclidean case, it is possible to estimate the score for non-Euclidean data using the trained autoencoders.

### 3.1    COORDINATE-INVARIANT GENERALIZATIONS OF THE AUTOENCODER COMPONENTS

Referring to Figure 2, let $\mathcal{M}$ be an $m$-dimensional manifold with local coordinates $x = (x^1, \ldots, x^m)$ and Riemannian metric $ds^2 = \sum_{i=1}^m \sum_{j=1}^m g_{ij}(x) \, dx^i dx^j$. Throughout this paper, we use italics to represent local coordinates, e.g., a point $\mathrm{x} \in \mathcal{M}$ has local coordinates $x \in \mathbb{R}^m$ and the mapping $\mathrm{r} : \mathcal{M} \to \mathcal{M}$ can be represented in local coordinates as $r : \mathbb{R}^m \to \mathbb{R}^m$. The metric will also be denoted in matrix form as $G(x) = (g_{ij}(x)) \in \mathbb{R}^{m \times m}$.

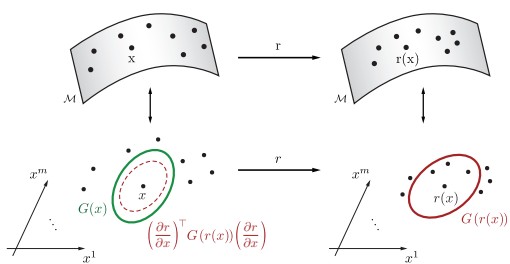

We now provide coordinate-invariant generalizations of each component in autoencoder training, especially in (1) and (2), while simultaneously reflecting any intrinsic properties of the manifold $\mathcal{M}$.

Figure 2:  Local coordinates and Riemannian metrics for the reconstruction mapping $\mathrm{r} : \mathcal{M} \to \mathcal{M}$. Local coordinates are denoted in italics.

**Reconstruction error:** The reconstruction error on Riemannian manifolds can be defined as $\text{dist}(r(x), x)^2$, where $\text{dist}(x, y)$ denotes the minimal geodesic distance between points $x, y \in \mathcal{M}$.

**Probability function:** Assume there exists a data-generating probability function $p_g : \mathcal{M} \to \mathbb{R}$ from which data points on $\mathcal{M}$ are drawn. Denote by $\rho_g : \mathbb{R}^m \to \mathbb{R}$ its representation in local coordinates, satisfying $\rho_g(x) > 0$ for all $x \in \mathbb{R}^m$ and $\int_{\mathcal{M}} \rho_g(x)\sqrt{\det G(x)}dx = 1$ (Pennec, 1999), where $\sqrt{\det G(x)}dx$ is the natural volume element induced from the metric. When formulating regularized autoencoders later, $p_g$ (or $\rho_g$) is used in the form of a weighted volume element $\rho(x)dx \equiv \rho_g(x)\sqrt{\det G(x)}dx$. In practice, the integrations involving $\rho(x)dx$ are approximated as an equally weighted finite sum of the integrands (with $\rho(x)$ excluded) evaluated at given data points.

**Contractive regularization:** The conventional Dirichlet energy (appearing in (2)) has been generalized to mappings between Riemannian manifolds in the theory of harmonic maps (Eells & Sampson, 1964), based on which we formulate the contractive regularization for non-Euclidean settings.

Let $\mathcal{M}$ be the input manifold, and let $\mathcal{N}$ be an $n$-dimensional output manifold with local coordinates $y = (y^1, \ldots, y^n)$ and Riemannian metric $dr^2 = \sum_{\alpha=1}^n \sum_{\beta=1}^n h_{\alpha\beta}(y) \, dy^\alpha dy^\beta$. The metric will also be denoted in matrix form as $H(y) = (h_{\alpha\beta}(y)) \in \mathbb{R}^{n \times n}$. In Eells & Sampson (1964) the Dirichlet energy of a smooth map $f : \mathcal{M} \to \mathcal{N}$ is defined as

$$\int_{\mathcal{M}} \text{Tr}(J^\top H J G^{-1})\sqrt{\det G} \, dx^1 \cdots dx^m, \tag{4}$$

where $J(x) = \left(\frac{\partial f^i}{\partial x^j}(x)\right) \in \mathbb{R}^{n \times m}$ is the differential $df_x : T_x\mathcal{M} \to T_y\mathcal{N}$ denoted in local coordinates. The energy functional in (4) is an intrinsic quantity, i.e., coordinate-invariant. We discuss the coordinate-invariance of (4) and some physical interpretations of minimizing (4) in Appendix B.1 and refer the reader to the extensive literature on the theory and applications of harmonic maps, e.g., Eells & Lemaire (1978; 1988); Park & Brockett (1994); Gu et al. (2004); Jang et al. (2021); Lee et al. (2021b).

To apply this energy functional to autoencoder regularization, we replace the mapping $f : \mathcal{M} \to \mathcal{N}$ and the Riemannian metric $H(f(x))$ with the reconstruction mapping $r : \mathcal{M} \to \mathcal{M}$ and $G(r(x))$, respectively, and replace the natural volume element $\sqrt{\det G} \, dx$ with the weighted volume element $\rho(x) \, dx$. Also note that for the case of non-Euclidean latent space $\mathcal{N}$, the objective in (4) with the volume element $\rho(x) \, dx$ can serve as a geometric regularizer for the contractive autoencoders. (See Appendix B.2 for more discussions related to non-Euclidean latent spaces.)

**Data corruption process:** To corrupt an input $x \in \mathcal{M}$, we sample a tangent vector $v \in T_x\mathcal{M}$ from an isotropic zero-mean multivariate Gaussian, i.e., a linear combination of an orthonormal basis for $T_x\mathcal{M}$ with coefficients sampled from $N(0, \sigma^2 I)$, and then apply the exponential map $\text{Exp}_x : T_x\mathcal{M} \to \mathcal{M}$ to $v$. The point $\tilde{x} = \text{Exp}_x(v)$ can then be interpreted as a corrupted point from $x$. We denote by $q(\tilde{x}|x)$ the noise density that samples $\tilde{x}$ for given $x$ according to this procedure. A data corruption example for data on a sphere ($S^2$) is illustrated in Figure 3, and a rationale for adopting this way of corruption is explained in Appendix B.3.

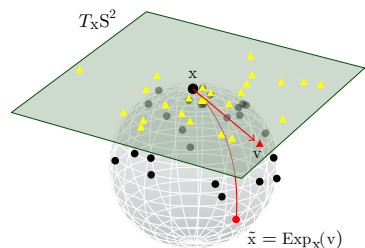

Figure 3: A data corruption example for an input $x \in S^2$. Tangent vectors (the yellow triangles) are sampled from a zero-mean Gaussian in $T_xS^2$ and mapped via the exponential map to black dots, the corrupted points from $x$. The red dot $\tilde{x} = \text{Exp}_x(v)$ for $v \in T_xS^2$.

### 3.2 GEOMETRICALLY REGULARIZED AUTOENCODERS

We now derive coordinate-invariant formulations of the regularized autoencoders presented in (1) and (2). For the reconstruction contractive autoencoder (2), the contractive regularizer modified from (4) (as explained above) is augmented to the reconstruction error term as follows:

$$\min_r \int_{\mathcal{M}} \left( \text{dist}(r(x), x)^2 + \sigma^2 \text{Tr}\left( \left(\frac{\partial r}{\partial x}\right)^\top G(r) \left(\frac{\partial r}{\partial x}\right) G^{-1} \right) \right) \rho(x) \, dx, \tag{5}$$

where the reconstruction map $r : \mathcal{M} \to \mathcal{M}$ is expressed in local coordinates as $r : \mathbb{R}^m \to \mathbb{R}^m$, and $G(r)$ denotes the metric at point $r(x)$. We refer to (5) as the geometric RCAE or GRCAE.

Similarly, the geometric version of the DAE (1), referred to as GDAE, can be formulated as follows:

$$\min_r \int_{\mathcal{M}} E_{q(\tilde{x}|x)} \left[ \text{dist}(r(\tilde{x}), x)^2 \right] \rho(x) \, dx, \tag{6}$$

where $E_{q(\tilde{x}|x)} [\, \cdot \,]$ denotes the expectation with respect to the noise density $q(\tilde{x}|x)$ presented above.

From the geometric formulations (5)-(6), we can obtain the following relations between the reconstruction function $r$ and the log of the probability function $\rho_g(x) = \frac{\rho(x)}{\sqrt{\det G(x)}}$.

**Theorem 1.** *Provided $\sigma^2$ is small, the derivative of the log of the probability function $\rho_g(x) = \frac{\rho(x)}{\sqrt{\det G(x)}}$ can be approximated for both the GRCAE and GDAE as*

$$\frac{\partial \log \rho_g(x)}{\partial x} = \frac{1}{\rho_g} \frac{\partial \rho_g}{\partial x}(x) = G(x) \left( \frac{r(x) - x}{\sigma^2} \right) + \mathcal{O}(\sigma^2). \tag{7}$$

The proof of Theorem 1 is provided in Appendix C. Equation (7) can be thought of as a generalization of (3) for non-Euclidean data, and we will refer to $\frac{\partial \log \rho_g(x)}{\partial x}$ as the *geometric score*.

The reconstruction function $r : \mathcal{M} \to \mathcal{M}$ for proposed autoencoders is modeled as a neural network in later experiments, with implementation details provided in Appendix D. We also provide there some ideas to deal with the case of manifolds that require multiple coordinate charts and discuss another case where the data space Riemannian metric is not known *a priori*.

## 4 EXPERIMENTS

In the experiments, we first demonstrate the geometric score estimation (Theorem 1) based on our geometrically regularized autoencoders for non-Euclidean data, providing a solid basis for future applications of autoencoders. We then utilize the proposed autoencoders for various applications, such as data sampling based on the Langevin Monte Carlo methods (Girolami & Calderhead, 2011) or clustering and noise filtering based on mode-seeking (Fukunaga & Hostetler, 1975; Cheng, 1995; Comaniciu & Meer, 2002), involving real-world non-Euclidean data sets. We also examine the usefulness of the proposed autoencoders in the representation learning perspective, using noisy point cloud data.

### 4.1 GEOMETRIC SCORE ESTIMATION

For geometric score estimation, we consider the data sampled from $P(n)$, the space of $n \times n$ symmetric positive-definite matrices, endowed with the affine-invariant Riemannian metric (Fletcher & Joshi, 2007). We train the GDAE and GRCAE using synthetic data sampled from $m$ mixtures of isotropic tangent space Gaussians for which the ground truth geometric score values are obtainable. For purposes of comparison, we also use DAE, RCAE, the least-squares log-density gradient method (LSLDG) presented in Sasaki et al. (2014), and also their extension to data on Riemannian manifolds (R-LSLDG) in Ashizawa et al. (2017). Full experimental details are provided in Appendix F.

For a given data set $\{x_1, \ldots, x_N\}$ represented in local coordinates of $P(n)$, the geometric score estimation error (Est. error), which is also defined to be coordinate-invariant, is evaluated as follows:

$$\text{Est. error} = \frac{1}{N} \sum_{i=1}^{N} \left( \frac{\partial \log \rho_g}{\partial x} \Big|_{\text{est}}(x_i) - \frac{\partial \log \rho_g}{\partial x}(x_i) \right)^\top G^{-1}(x_i) \left( \frac{\partial \log \rho_g}{\partial x} \Big|_{\text{est}}(x_i) - \frac{\partial \log \rho_g}{\partial x}(x_i) \right), \tag{8}$$

where $\frac{\partial \log \rho_g}{\partial x} \Big|_{\text{est}}(x_i)$ and $\frac{\partial \log \rho_g}{\partial x}(x_i)$ respectively denote the estimated value and the ground truth value for $\frac{\partial \log \rho_g(x)}{\partial x}$ at $x_i$. The estimation errors measured on $10,000$ test data points are averaged over five runs in Table 1. Note that the GDAE, GRCAE, and R-LSLDG methods show superior performance over DAE, RCAE, and LSLDG in estimating $\frac{\partial \log \rho_g(x)}{\partial x}$. Notably, GRCAE shows the best performance for higher dimensionality and a higher number of mixtures in terms of estimation error. We also obtain a similar tendency for another synthetic data set on the hypersphere $S^n = \{p \in \mathbb{R}^{n+1} \mid \|p\| = 1\}$, and the results are provided in Appendix F.4.

Table 1: Estimation errors for $\frac{\partial \log \rho_g}{\partial x}$ for $m$ mixtures of tangent space Gaussian data on P($n$) with the standard errors in parentheses. Bolds represent the best and comparable methods from the t-test with a significance level of 5%.

| $n$ | $m$ | LSLDG | DAE | RCAE | R-LSLDG | GDAE (ours) | GRCAE (ours) |
|---|---|---|---|---|---|---|---|
| 2 | 2 | 10.5 (0.87) | 8.72 (3.03) | 9.04 (2.49) | 3.84 (0.19) | 3.55 (0.31) | **2.61** (0.20) |
| 3 | 2 | 30.0 (0.91) | 17.9 (1.23) | 18.1 (1.83) | 5.64 (0.19) | 4.95 (0.27) | **4.33** (0.34) |
| 3 | 3 | 37.6 (2.16) | 21.0 (0.97) | 22.4 (1.18) | **7.07** (0.31) | **6.86** (0.96) | **5.85** (1.07) |
| 4 | 2 | 75.5 (0.85) | 42.4 (0.69) | 43.8 (1.01) | 9.67 (0.78) | **7.72** (0.60) | **6.97** (0.39) |
| 4 | 3 | 90.8 (1.91) | 59.5 (4.41) | 63.3 (6.25) | **12.2** (1.45) | **11.8** (0.71) | **10.5** (1.06) |
| 4 | 4 | 88.4 (0.82) | 73.2 (2.98) | 78.1 (4.76) | 13.9 (0.82) | 16.0 (1.29) | **12.6** (0.28) |

## 4.2 SAMPLING ON $S^2$

We can apply the geometric scores estimated from GDAE and GRCAE to the sampling of non-Euclidean data via Riemannian Langevin Monte Carlo (RLMC) methods. The stochastic process for the RLMC methods in Girolami & Calderhead (2011) can be reformulated using the geometric score in (7) as follows:

$$dx = \frac{1}{2}\left(G^{-1}(x)\frac{\partial \log \rho_g(x)}{\partial x} - \frac{1}{\beta}G^{-1}(x)\frac{\partial \beta}{\partial x} + \Psi(x)\right)dt + \sqrt{G^{-1}(x)}dw, \qquad (9)$$

where $dw \in \mathbb{R}^m$ denotes the Brownian motion in an $m$-dimensional vector space, $\beta(x) = 1/\sqrt{\det G(x)}$, and $\Psi(x) \in \mathbb{R}^m$ is a vector whose $i$-th component is given by $\sum_{j=1}^{m}\frac{\partial g^{ij}}{\partial x^j}$ with $g^{ij}$ as the $(i, j)$ element of $G^{-1}(x)$. A discretization of the above process gives the RLMC method (Girolami & Calderhead, 2011). Note that in (9), the terms except for $\frac{1}{2}G^{-1}(x)\frac{\partial \log \rho_g(x)}{\partial x}dt$ correspond to the Brownian motion on manifolds (Brockett, 1997).

As an illustrative case study, we consider sampling on a sphere ($S^2$). After training DAE, RCAE, GDAE, and GRCAE for data points on $S^2$ shown in Figure 4 (left), we sample new data points that approximately follow the original data distribution by applying the RLMC methods using the geometric scores estimated from GDAE/GRCAE and its Euclidean counterparts using DAE/RCAE. We also report the results obtained from the S-Flow method (a variation of the M-Flow method (Brehmer & Cranmer, 2020) for the case of $S^2$). The experimental details are provided in Appendix G.

For a quantitative comparison of the sampling performances, the maximum mean discrepancy (MMD) (Gretton et al., 2012) between a test data set and the obtained samples is provided in Table 2. As shown in the table and Figure 4, GDAE shows the best quantitative and qualitative performance among the considered autoencoders. Even though the samples from GDAE do not yield better numerical results than those from the S-Flow method, an algorithm targeted for data sampling, it is observed that plausible samples can be obtained. Also note that the samples obtained from RCAE and GRCAE tend to be inferior to those from DAE and GDAE in this task, possibly due to the algorithmic properties of RCAE/GRCAE in which, unlike DAE/GDAE, the inputs to neural networks are strictly confined to the given training data. Thus, the reconstruction functions may not be accurately trained on other regions visited during the sampling process compared to DAE/GDAE.

Table 2: The MMD measure (the lower, the better) between the test and the sampled data with standard errors in parentheses.

| $\times 10^{-2}$ | DAE | RCAE | GDAE (ours) | GRCAE (ours) | S-Flow |
|---|---|---|---|---|---|
| four blobs | 0.215 (0.015) | 0.598 (0.059) | **0.188** (0.018) | 0.291 (0.052) | **0.137** (0.038) |
| two moons | 0.147 (0.022) | 0.249 (0.020) | **0.144** (0.032) | 0.161 (0.020) | **0.100** (0.030) |
| s-curve | 0.300 (0.042) | 0.248 (0.048) | **0.242** (0.031) | **0.231** (0.072) | 0.326 (0.063) |
| circles | 0.075 (0.021) | 0.353 (0.012) | **0.061** (0.005) | 0.323 (0.008) | **0.046** (0.093) |

## 4.3 CLUSTERING OF HYPERSPHERICAL DATA

In natural language processing, the similarity between word embeddings, i.e., the vector representations that encode semantic information of the words, is often measured according to the cosine

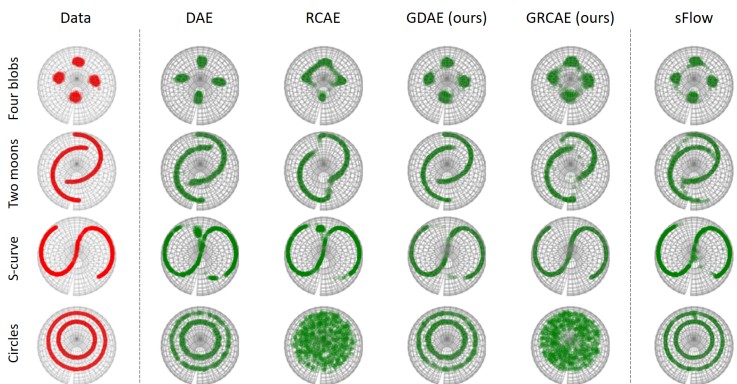

Figure 4: Data sampling on $S^2$.

similarity (Mikolov et al., 2013). This can be equated to considering the embeddings as points on a hypersphere and measuring the distance based on the geodesic distance (Straub et al., 2015).

Based on this idea, we group documents in the Newsgroup20 data set (Lang, 1995) using the GDAE trained on the document embeddings. The document embeddings are first represented as the average of the word embeddings in the document and then projected to a hypersphere. For the word embeddings, we utilize pre-trained 50-dimensional GloVe embeddings (Pennington et al., 2014), hence the document embeddings lie on $S^{49}$. We define four clustering tasks as described in Appendix H.1.

We train DAE, GDAE, LSLDG, and R-LSLDG as explained in Appendix H.2. Here we consider two variations of DAE and LSLDG, respectively; the first ones are trained on the spherical coordinate representations of the data, and the second ones on the representations in the ambient space $\mathbb{R}^{n+1}$. After training the models, the document embeddings are iteratively updated along the gradient of the log-probability (for GDAE, this can be performed by repeatedly applying the reconstruction function on the embedding) for a fixed number of steps and then grouped as done in the mean shift clustering (Comaniciu & Meer, 2002). The adjusted Rand index (ARI) is used as the performance metric for the clustering tasks (Hubert & Arabie, 1985).

Table 3: The adjusted Rand index for the clustering results on the Newsgroup20 data set ($S^{49}$) with standard errors in parentheses. The higher, the better.

| Tasks | LSLDG | LSLDG Ambient | DAE | DAE Ambient | R-LSLDG | GDAE (ours) |
|---|---|---|---|---|---|---|
| crypt | 0.174 (0.016) | 0.034 (0.025) | 0.205 (0.069) | 0.071 (0.040) | 0.221 (0.109) | **0.465** (0.018) |
| electronics | 0.204 (0.006) | 0.036 (0.018) | 0.228 (0.082) | 0.092 (0.001) | 0.192 (0.099) | **0.448** (0.044) |
| med | 0.153 (0.012) | 0.021 (0.017) | 0.227 (0.120) | 0.071 (0.040) | 0.246 (0.090) | **0.451** (0.055) |
| space | 0.176 (0.019) | 0.025 (0.026) | 0.211 (0.066) | 0.089 (0.001) | 0.241 (0.081) | **0.429** (0.070) |

The averaged clustering performance for five runs of each method is presented in Table 3. Note that the vector-spaced methods trained on the ambient representations of the data are hardly successful in grouping the data. On average, the R-LSLDG performs slightly better than LSLDG and DAE but with higher variance. The results obtained from GDAE show a much higher ARI than others. An additional case study for the clustering of covariance matrix data is provided in Appendix H.3.

## 4.4    FILTERING OF DIFFUSION TENSOR IMAGING DATA

For the next case study, we consider data obtained from diffusion tensor imaging (DTI). Mathematically, a DTI datum is a three-dimensional image in which the value assigned to each voxel is an element of P(3). A voxel of DTI data can be treated as in the space of three-dimensional normal distributions N(3), by regarding the voxel location and voxel value as

Table 4: The R-squared score for N(2) data filtering. The higher, the better.

| Noise level | DAE | GDAE (ours) | MVKR |
|---|---|---|---|
| 0.02 | 99.80 (0.00) | **99.86** (0.00) | 96.31 (0.01) |
| 0.05 | 99.12 (0.05) | **99.51** (0.03) | 96.21 (0.03) |
| 0.1 | 97.14 (0.16) | **98.51** (0.03) | 95.79 (0.04) |
| 0.2 | 89.44 (0.32) | **93.70** (0.47) | **94.11** (0.10) |

the mean and covariance of a normal distribution, respectively. In this section, following Han & Park (2014), we adopt the Fisher information Riemannian metric (FIRM) for DTI data (see Appendix D.3.1 for the definition of the FIRM). We train the GDAE on raw DTI data and apply the trained GDAE to filter the noise in the data.

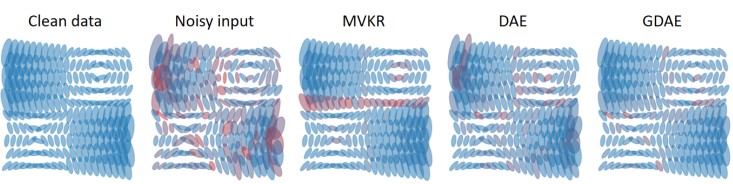

Figure 5: N(2) data filtering (noise 0.2). The redder, the higher error.

Consider a data set comprised of $N$ voxels $\{(x_1, P_1), \ldots, (x_N, P_N)\}$ of a raw DTI datum, where $x_i \in \mathbb{R}^3$ and $P_i \in \mathrm{P}(3)$ denote the location and value of the $i$-th voxel, respectively. For the data set, we train the GDAE with the reconstruction function r : N(3) → N(3); the overall training process is explained in Appendix I.1.

By iteratively applying the reconstruction function of the trained GDAE, data points are mapped toward the local modes of the probability function (as can be implied from (7)), and the voxels with added noise, which usually have a lower probability, can be automatically filtered (Comaniciu & Meer, 2002). We summarize a DTI filtering algorithm based on the GDAE in Appendix I.2.

We first demonstrate the effectiveness of this algorithm in a simpler setting by using a synthetic data set in two-dimensional normal distributions N(2). We train our GDAE on a noisy input artificially corrupted from clean data in Figure 5 and apply the filtering algorithm. The proposed algorithm can effectively filter out the noises qualitatively better than other filtering methods, such as the DAE-based filtering with more remaining noises and a manifold-valued kernel regression-based filtering approach (MVKR) (Banerjee et al., 2015) which tends to erroneously smooth discontinuous voxel values. A quantitative comparison in Table 4 also suggests that our GDAE-based filtering can perform well at various noise levels. The experimental details are deferred to Appendix I.3.

We now conduct numerical experiments of DTI filtering by applying our algorithm. Data used in these experiments were obtained from the Alzheimer's Disease Neuroimaging Initiative (ADNI) database (adni.loni.usc.edu). See Appendix I.4 for the data preprocessing details. We also perform filtering based on DAE trained on nine-dimensional vector representations of voxels. Note that, critically, the DAE-based filtering has no guarantee for the filtered voxel values to be positive-definite, while GDAE-based filtering always satisfies the constraint.

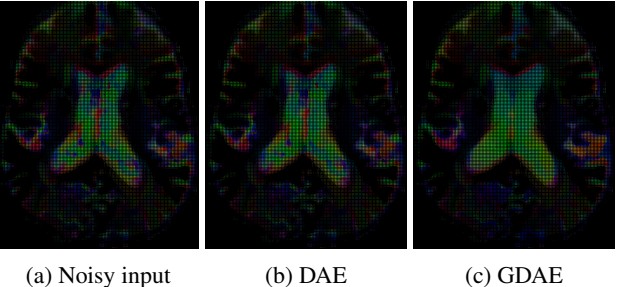

(a) Noisy input     (b) DAE     (c) GDAE

Figure 6: DTI filtering results.

The input image and the filtering results from each algorithm are shown in Figure 6. We plot axial slices of the corresponding DTI data in Figures 6 (a)-(c). Voxel values are drawn as ellipsoids with colors representing the direction of the first principal axis.

Since the ground truths of DTI data are not available, only qualitative comparisons between the filtering results can be made. In Figures 6 (a)-(c), it can be seen that the GDAE-based method effectively filters outliers (the abruptly changing colors appearing in Figure 6 (a)) when compared to DAE-based filtering. Using brain anatomical terms, the separation between the cerebrospinal fluid and brain parenchyma became clear after GDAE-based filtering, and as a result specifying the sulci and gyri of the brain is much easier than from the raw input or DAE-based filtering results. Furthermore, noise in the ventricle area also disappeared after GDAE-based filtering, while DAE-based filtering failed to eliminate this noise. We should note here that the DTI filtering results would require further concordance verification procedures with experts such as radiologists to ensure that only spurious artifacts are erased rather than important anatomy.

## 4.5 Robust Representation Learning of Point Cloud Data

We now show that we can obtain useful representations for non-Euclidean data from our proposed autoencoders, especially when noise exists in the data. We consider point cloud data in this case study. A point cloud data in $\mathbb{R}^D$ is a set of $n$ points in $\mathbb{R}^D$, represented as $X = \{x_1, \ldots, x_n \mid x_i \in \mathbb{R}^D\}$. Recently, a statistical manifold framework has been suggested for point cloud data in Lee et al. (2022), and they have observed that reflecting this geometry to train autoencoders can obtain better representations than the vector space counterparts. (See Appendix D.4.1 for more details.)

Based on this choice of geometry, we train GDAE for point cloud data with $n = 2,048$ and $D = 3$ (hence of dimension $nD = 6,144$). We reflect the Fisher information metric proposed in Lee et al. (2022) in the data corruption process for GDAE and use the modified Chamfer loss as the reconstruction error for point cloud data. For comparison purposes, we also consider DAE trained on the vector representations of point cloud data. We use the FcNet (Yang et al., 2018) as the structure for the reconstruction functions with a latent space dimension of 512. Further details are provided in Appendix D.4.

To verify the usefulness of the representations obtained from our trained autoencoders, we utilize them as features to train classifiers in the transfer learning setting based on some benchmark data sets following Yang et al. (2018). More specifically, we train autoencoders using the ShapeNet data set (Chang et al., 2015) and obtain representations for the ModelNet data set (Wu et al., 2015). We then train a linear SVM using the representations and measure the transfer classification accuracy. We inject varying noises in the data sets to verify if the obtained representations are robust to noise. More experimental details are explained in Appendix J.

Table 5: Classification accuracy by transfer learning for ModelNet10 and ModelNet40 from ShapeNet under the noise levels of 0.01, 0.05, 0.1, and 0.2.

| Noise level | ModelNet10 | | | | | ModelNet40 | | | | |
| --- | --- | --- | --- | --- | --- | --- | --- | --- | --- | --- |
| | AE | DAE | GDAE (ours) | AE +R. | GDAE (ours) + R. | AE | DAE | GDAE (ours) | AE + R. | GDAE (ours) + R. |
| 0.01 | 93.1 | 92.7 | **93.2** | 92.8 | **93.2** | 87.3 | 87.0 | 87.3 | **88.6** | **88.8** |
| 0.05 | 91.3 | 91.6 | 92.3 | **93.3** | **92.4** | 83.2 | 84.6 | 85.0 | **86.8** | **86.5** |
| 0.1 | 88.9 | 90.6 | **91.1** | 90.2 | **91.7** | 75.6 | 79.9 | **83.2** | 80.5 | **82.4** |
| 0.2 | 79.8 | 81.2 | **88.2** | 84.8 | **88.0** | 64.6 | 67.3 | **77.3** | 72.2 | **76.1** |

The experimental results are in Table 5. The features obtained from GDAE lead to a better transfer classification accuracy than those from the vanilla autoencoder (AE) or DAE; this tendency gets stronger as the noise level increases. Our approach also shows comparable or better performances compared to another regularization method ('AE + R.' in Table 5) considered in Lee et al. (2022), which tries to match the pull-back metric of the Fisher information metric (via the decoder mapping) to the identity. Combining these two regularization methods ('GDAE + R.' in Table 5) shows the best overall transfer classification accuracy, demonstrating the usefulness of our geometric regularization methods in obtaining better representations of non-Euclidean data for downstream tasks.

## 5 Conclusion

In this paper, we have introduced geometrically regularized autoencoders for non-Euclidean data. By constructing regularization terms in a coordinate-invariant way, we have developed two types of geometric autoencoders, the geometric reconstruction contractive autoencoder and the geometric denoising autoencoder. These autoencoders are effective in estimating the derivative of the log of the probability density of non-Euclidean data and have been successfully applied to several applications such as data sampling, mode-seeking, and representation learning tasks involving real-world non-Euclidean data sets. Although training these models can be computationally demanding and may involve numerical issues when reflecting the geometry of the data, our experiments show that our approach can be a viable option for handling high-dimensional and complex non-Euclidean data. In the future, it would be worth investigating more efficient and robust methods for reflecting the geometry during training. Additionally, the idea of using geometric regularizations could be applied to other machine learning problems that involve non-Euclidean data.

ACKNOWLEDGMENTS

C. Jang and Y.-K. Noh were supported by IITP Artificial Intelligence Graduate School Program for Hanyang University funded by MSIT (Grant No. 2020-0-01373). Y.-K. Noh was partly supported by NRF/MSIT (Grant No. 2018R1A5A7059549, 2021M3E5D2A01019545) and IITP/MSIT (Grant No. IITP-2021-0-02068). Y. Lee and F. C. Park were supported in part by SRRC NRF grant 2016R1A5A1938472, IITP-MSIT (Grant No. 2022-0-00480, Development of Training and Inference Methods for Goal-Oriented AI Agents, 20%), SNU-AIIS, SNU-IAMD, and the SNU Institute for Engineering Research.

REPRODUCIBILITY STATEMENT

We refer the reader to the following pointers for reproducibility:

- Codes to train the proposed autoencoders: Supplementary Material.
- Proof of Theorem 1: Appendix C.
- Implementation details of the proposed autoencoders: Appendix D.
- Experimental settings: Appendix E, F, G, H, I, and J.

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

APPENDIX

## A   MATHEMATICAL BACKGROUNDS

In this section, we briefly review some notions related to differentiable manifolds and Riemannian geometry. For further mathematical details on differentiable manifolds and Riemannian geometry, we refer the reader to Boothby (1986) and Dubrovin et al. (1992).

Intuitively, an $m$-dimensional differentiable manifold $\mathcal{M}$ is a space which is locally diffeomorphic[1] to $m$-dimensional Euclidean space. For every point p $\in \mathcal{M}$, there exists a coordinate chart $(U, x)$, where $U$ is an open subset of $\mathcal{M}$ containing p, and $x$ is a homeomorphism[2] of $U$ to an open subset of $\mathbb{R}^m$. Applying $x$ to p gives the $m$ coordinates of p, i.e., $x(\text{p}) = (x^1(\text{p}), \ldots, x^m(\text{p})) \in \mathbb{R}^m$—each $x^i$ is a real-valued function on $U$, the $i$-th coordinate function. Here $x$ is called the local coordinates; note that other choices of local coordinates are also possible (e.g., for the sphere, both spherical coordinates and stereographic projection correspond to different local coordinates of the sphere).

A differentiable manifold $\mathcal{M}$ endowed with a Riemannian metric is called a Riemannian manifold. The Riemannian metric is a function defined on the manifold $\mathcal{M}$ that assigns to each point p $\in \mathcal{M}$ a bilinear mapping $\Phi_\text{p} : T_\text{p}\mathcal{M} \times T_\text{p}\mathcal{M} \to \mathbb{R}$, where $T_\text{p}\mathcal{M}$ denotes the tangent space to $\mathcal{M}$ at p. Using the local coordinates $x = (x^1, \ldots, x^m)$, the Riemannian metric can be expressed as $\Phi = \sum_{i=1}^{m} \sum_{j=1}^{m} g_{ij}(x)dx^i dx^j$ or $ds^2 = \sum_{i=1}^{m} \sum_{j=1}^{m} g_{ij}(x)dx^i dx^j$. Here $g_{ij}(x)$ is assumed to be smooth, i.e., infinitely differentiable, and its matrix representation $G = (g_{ij}) \in \mathbb{R}^{m \times m}$ is symmetric positive-definite. The Riemannian metric allows one to calculate lengths, angles, volumes, and even define a distance metric on differentiable manifolds in an intrinsic way, i.e., in a way that is invariant to the choice of local coordinates.[3]

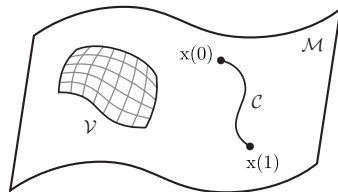 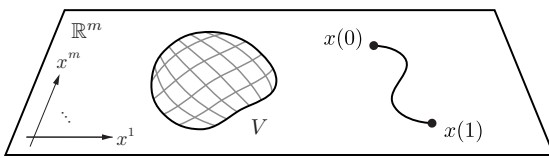

(a) A curve and a region on $\mathcal{M}$          (b) Corresponding curve and region in local coordinates

Figure 7: A Riemannian manifold $\mathcal{M}$ and its local coordinate $x$.

The length of a curve $\mathcal{C} = \{\text{x}(t) \in \mathcal{M} \mid t \in [0, 1]\}$ is calculated as

$$\text{Length}(\mathcal{C}) = \int_0^1 \sqrt{\dot{x}(t)^\top G(x(t))\dot{x}(t)} \, dt, \tag{10}$$

where $x(t) \in \mathbb{R}^m$ is the local coordinate representation of x$(t)$ (see Figure 7). Given two fixed boundary points x$(0)$, x$(1) \in \mathcal{M}$, the curves that minimize the length (10) are called the minimal geodesics, and the corresponding lengths the minimal geodesic distances.

The equations for geodesics are obtained as

$$\frac{d^2}{dt^2}x^k + \sum_{i=1}^{m} \sum_{j=1}^{m} \Gamma_{ij}^k \frac{dx^i}{dt} \frac{dx^j}{dt} = 0, \quad k = 1, \ldots, m, \tag{11}$$

where $\Gamma_{ij}^k$ denote the Christoffel symbols of the second kind in $\mathcal{M}$, i.e.,

$$\Gamma_{ij}^k = \sum_{l=1}^{m} \frac{1}{2} g^{kl} \left( \frac{\partial g_{li}}{\partial x^j} + \frac{\partial g_{lj}}{\partial x^i} - \frac{\partial g_{ij}}{\partial x^l} \right), \tag{12}$$

and $g^{kl}$ is the $(k, l)$ entry of $G^{-1} \in \mathbb{R}^{m \times m}$.

---

[1]Two manifolds are said to be diffeomorphic if there exists a differentiable mapping between the two manifolds which is invertible and its inverse is also differentiable. Such a mapping is called a diffeomorphism.

[2]A continuous function is called a homeomorphism if it is invertible, and its inverse is continuous.

[3]See Appendix B.1 for more discussions on the coordinate transformations and the coordinate-invariance.

Consider a geodesic curve x : $[0,1] \to \mathcal{M}$ emanating from a point x(0) $\in \mathcal{M}$ with an initial velocity vector $\dot{x}(0) = v \in T_{x(0)}\mathcal{M}$ as shown in Figure 8. It is known that such a geodesic is unique if it exists (Boothby, 1986), and denote by x(1) $\in \mathcal{M}$ the endpoint of the geodesic that propagates for a unit time. The mapping that maps the initial velocity vector v to the point x(1) is called the exponential map $\text{Exp}_{x(0)} : T_{x(0)}\mathcal{M} \to \mathcal{M}$. Note that the distance between x(0) and x(1) is the same as the norm of v. This corresponds to a generalization of propagating a line in vector space from a starting point along a vector.

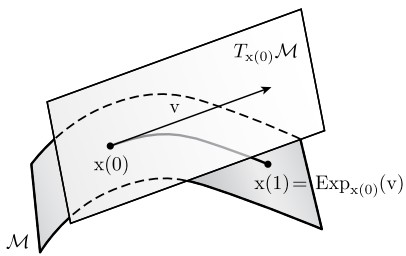

Figure 8: A geodesic curve x : $[0,1] \to \mathcal{M}$ emanating from x(0) $\in \mathcal{M}$ along $v \in T_{x(0)}\mathcal{M}$.

On Riemannian manifolds, there exists a (natural) volume element induced from the Riemannian metric $G(x)$ which is expressed in local coordinates as $\sqrt{\det G(x)}\, dx^1 \cdots dx^m$. The volume of a compact subset $\mathcal{V} \subseteq \mathcal{M}$ is then obtained by the following integral:

$$\text{Volume}(\mathcal{V}) = \int_V \sqrt{\det G(x)}\, dx^1 \cdots dx^m, \tag{13}$$

where $V$ denotes the domain of integration expressed in local coordinates[4] (see Figure 7). The integration of a bounded and continuous function $f : \mathcal{M} \to \mathbb{R}$ on the integration domain $\mathcal{V}$ is also obtained using the volume element as follows:

$$\int_V f(x)\sqrt{\det G(x)}\, dx^1 \cdots dx^m. \tag{14}$$

# B FURTHER EXPLANATIONS OF THE GEOMETRIC REGULARIZATION COMPONENTS

## B.1 CONTRACTIVE REGULARIZATION

To see why (4) is coordinate-invariant, observe that under a pair of local coordinate transformations $x \mapsto x' = \phi(x)$ and $y \mapsto y' = \psi(y)$, $G(x) = (g_{ij}(x)) \in \mathbb{R}^{m \times m}$, $H(y) = (h_{\alpha\beta}(y)) \in \mathbb{R}^{n \times n}$, and $J(x) = \left(\frac{\partial f^i}{\partial x^j}(x)\right) \in \mathbb{R}^{n \times m}$ transform according to the following rules (Dubrovin et al., 1992): (i) $G \mapsto G' = \Phi^{-\top} G \Phi^{-1}$, where $\Phi = \frac{\partial \phi}{\partial x} \in \mathbb{R}^{m \times m}$; (ii) $H \mapsto H' = \Psi^{-\top} H \Psi^{-1}$, where $\Psi = \frac{\partial \psi}{\partial y} \in \mathbb{R}^{n \times n}$; (iii) $J \mapsto J' = \Psi J \Phi^{-1}$,[5] where it can be verified that $\text{Tr}\left(J^\top H J G^{-1}\right)$ remains the same.

Also note that minimizing (4) induces the contraction (shrinking without distortion) of the mapping f. As an extreme case without any boundary conditions or constraints, trivial solutions are obtained as $J = 0$ or equivalently f = constant, which is an extreme contraction. On the other hand, provided the boundary conditions or constraints for f are well-specified, a useful physical analogy for minimizing (4) is to imagine wrapping a curved object made of marble ($\mathcal{N}$) by an elastic sheet ($\mathcal{M}$); harmonic maps, which are extrema of (4), can be viewed as solutions corresponding to elastic equilibria (Eells & Sampson, 1964).

## B.2 SOME REMARKS ON NON-EUCLIDEAN LATENT SPACES

Recently, to better capture the structure of data distributions, there have been increasing works on autoencoders that deal with non-Euclidean latent spaces, e.g., hyperspheres (Davidson et al., 2018; Xu & Durrett, 2018), hyperbolic spaces or Poincaré balls (Mathieu et al., 2019), their mixtures

---

[4]We may also use $\mathcal{V}$ rather than $V$ to denote the domain of integration for notational simplicity.

[5]We can easily verify (i) and (ii) by observing the Riemannian metric should remain the same under the local coordinate transform as $ds^2 = [dx]^\top G[dx] = [dx']^\top G'[dx']$, where $[dx] = [dx^1, \ldots, dx^m]^\top$ and $[dx'] = [dx'^1, \ldots dx'^m]^\top$ are related by $[dx] = \frac{\partial x}{\partial x'}[dx'] = \Phi^{-1}[dx']$. We can verify (iii) by considering the chain rule $\frac{\partial(y' \circ f)}{\partial x'} = \frac{\partial y'}{\partial y}\frac{\partial f}{\partial x}\frac{\partial x}{\partial x'} = \Psi\frac{\partial f}{\partial x}\Phi^{-1}$.

(Skopek et al., 2019), or submanifolds embedded in Euclidean spaces (Rey et al., 2019). In the case of such non-Euclidean latent spaces, by reflecting the coordinate-invariant contractive regularization discussed in Section 3.1, we can formulate geometric contractive autoencoder (GCAE) as follows:

$$\min_{r=g\circ f} \int_{\mathcal{M}} \left( \text{dist}(\text{r(x)}, \text{x})^2 + \sigma^2 \, \text{Tr} \left( \left( \frac{\partial f}{\partial x} \right)^{\top} H(f(x)) \left( \frac{\partial f}{\partial x} \right) G^{-1} \right) \right) \rho(x) \, dx, \qquad (15)$$

where the reconstruction map r $: \mathcal{M} \to \mathcal{M}$, the encoder mapping f $: \mathcal{M} \to \mathcal{N}$, and the decoder mapping g $: \mathcal{N} \to \mathcal{M}$ are respectively expressed in local coordinates as $r : \mathbb{R}^m \to \mathbb{R}^m$, $f : \mathbb{R}^m \to \mathbb{R}^n$, and $g : \mathbb{R}^n \to \mathbb{R}^m$.

Training the GCAE may induce additional regularization effects that better capture data-concentrated regions or make the model more robust to noise, in addition to the effect of using non-Euclidean latent spaces. Confirming these experimentally is left for future work.

As another choice of the Riemannian metric for latent space, we can consider the pullback of the data space metric via decoder mapping (Arvanitidis et al., 2018). It has been observed that applying this metric can better characterize the data distances in latent space and provide more meaningful results in analyzing latent representations. Adopting this metric on the latent space reveals an interesting connection between the GCAE in (15) and GRCAE in (5). This pullback metric is obtained as $H(y) = \left( \frac{\partial g}{\partial y} \right)^{\top} G(g(y)) \left( \frac{\partial g}{\partial y} \right) \in \mathbb{R}^{n \times n}$, and we can observe that (15) and (5) become identical when substituting this metric into (15) and considering the chain rule $\frac{\partial r}{\partial x} = \frac{\partial g}{\partial y} \frac{\partial f}{\partial x} \in \mathbb{R}^{m \times m}$.

### B.3 DATA CORRUPTION PROCESS

The corruption process $\tilde{x} \sim N(x, \sigma^2 I)$ in vector space is equivalent to set $\tilde{x} = x + \epsilon$ for $\epsilon \sim N(0, \sigma^2 I)$, i.e., the endpoint of a line segment starting from $x$ and extended according to the vector $\epsilon$. On the Riemannian manifolds, a similar discussion is possible using the notion of the geodesic and the exponential map as discussed in Appendix A. Therefore, we corrupt an input x $\in \mathcal{M}$ by applying the exponential map $\text{Exp}_{\text{x}} : T_{\text{x}}\mathcal{M} \to \mathcal{M}$ to a tangent vector v $\in T_{\text{x}}\mathcal{M}$ sampled from an isotropic zero-mean multivariate Gaussian, i.e., v $= v^1 E_1 + \cdots + v^m E_m$ for an orthonormal basis $E_1, \ldots, E_m$ for $T_{\text{x}}\mathcal{M}$ and an $m$-dimensional vector $(v^1, \ldots, v^m) \sim N(0, \sigma^2 I)$. By using an isotropic Gaussian, the corruption process $\tilde{\text{x}} = \text{Exp}_{\text{x}}(\text{v})$ does not depend on which orthonormal basis $E_1, \ldots, E_m$ is used.

## C  PROOF OF THEOREM 1

### C.1  THE FIRST-ORDER NECESSARY CONDITIONS FOR GRCAE

**Lemma 1.** *Provided $\sigma^2$ is small, the derivative of the log of the probability function $\rho_g(x) = \frac{\rho(x)}{\sqrt{\det G(x)}}$ can be approximated for the geometric reconstruction contractive autoencoder as*

$$\frac{\partial \log \rho_g(x)}{\partial x} = \frac{1}{\rho_g} \frac{\partial \rho_g}{\partial x}(x) = \frac{1}{\rho} \frac{\partial \rho}{\partial x}(x) - \Gamma(x) = G \left( \frac{r(x) - x}{\sigma^2} \right) + \mathcal{O}(\sigma^2), \qquad (16)$$

*where $\Gamma(x) \in \mathbb{R}^m$ is a vector whose $i$-th component is given by $\frac{1}{2}\text{Tr} \left( G^{-1} \frac{\partial G}{\partial x^i} \right)$.*

*Proof.* The first-order necessary conditions for (5) can be obtained from the following Euler-Lagrange equations:

$$\frac{\partial \mathcal{L}}{\partial r^i} - \sum_{j=1}^{m} \frac{\partial}{\partial x^j} \left( \frac{\partial \mathcal{L}}{\partial \left( \frac{\partial r^i}{\partial x^j} \right)} \right) = 0, \qquad i = 1, \ldots, m, \qquad (17)$$

where $\mathcal{L}$ is the integrand of (5) with an approximation on the squared geodesic distance to $(r(x) - x)^{\top} G(x)(r(x) - x)$ provided $\sigma^2$ small,[6] and $r^i$ denotes the $i$-th coordinate representation of the

---

[6]The difference between optimal $r(x)$ and $x$ turns out to be $\mathcal{O}(\sigma^2)$. It can be verified that this approximation does not affect the final results, since the approximation error of the squared geodesic distance becomes $\mathcal{O}(\sigma^6)$.

reconstruction function $r$. By applying $\mathcal{L}$ to the Euler-Lagrange equations, (17) results in

$$r^i - x^i = \sigma^2 \, \eta^i(x, r, r', r''), \qquad i = 1, \ldots, m, \tag{18}$$

where $r'$ denotes $\frac{\partial r}{\partial x}$, $r''$ denotes the collection of $\frac{\partial^2 r^j}{\partial x^2}$ for $j = 1, \ldots, m$, and $\eta^i$ denotes a function of $x, r, r', r''$.[7] Assuming $r, G, G^{-1}, \rho$ are smooth, and their higher-order derivatives are bounded, $r^i - x^i = \sigma^2 \eta^i(x, r, r', r'') = \mathcal{O}(\sigma^2)$ holds. By iterating the relation $r^i = x^i + \sigma^2 \eta^i(x, r, r', r'') = x^i + \mathcal{O}(\sigma^2)$, i.e., substituting this form of $r^i$ into $\eta^i(x, r, r', r'')$, the $i$-th component of $r$ is obtained for small $\sigma^2$ as follows:

$$r^i - x^i = \sigma^2 \left( \sum_{j=1}^{m} g^{ij} \left( \frac{1}{\rho} \frac{\partial \rho}{\partial x^j} - \frac{1}{2} \mathrm{Tr}\left( G^{-1} \frac{\partial G}{\partial x^j} \right) \right) \right) + \mathcal{O}(\sigma^4), \tag{19}$$

$$= \sigma^2 \left( \sum_{j=1}^{m} g^{ij} \frac{1}{\rho_g} \frac{\partial \rho_g}{\partial x^j} \right) + \mathcal{O}(\sigma^4), \tag{20}$$

where $g^{ij}$ denotes the $(i, j)$ entry of $G^{-1}$, $\rho_g = \frac{\rho}{\sqrt{\det G}}$, and the identity $\frac{\partial \log(\det G)}{\partial x^j} = \mathrm{Tr}\left( G^{-1} \frac{\partial G}{\partial x^j} \right)$ is used to derive (20). By gathering (20) for all $i$, the first-order necessary conditions can be rewritten as

$$\frac{1}{\rho_g} \frac{\partial \rho_g}{\partial x}(x) = G(x) \left( \frac{r(x) - x}{\sigma^2} \right) + \mathcal{O}(\sigma^2). \tag{21}$$

$\square$

## C.2 THE FIRST-ORDER NECESSARY CONDITIONS FOR GDAE

To obtain the first-order necessary conditions for GDAE, the integrand of (6) is first represented in local coordinates $x = (x^1, \ldots, x^m)$ in Lemma 2.

**Lemma 2.** *Provided $\sigma^2$ is small, $E_{q(\tilde{x}|x)}\left[ \mathrm{dist}(r(\tilde{x}), x)^2 \right]$ in (6) can be approximated in local coordinates as*

$$E_{q(\tilde{x}|x)}\left[ \mathrm{dist}(r(\tilde{x}), x)^2 \right] \tag{22}$$

$$= (r(x) - x)^\top G(x)(r(x) - x) + \sigma^2 \left( \mathrm{Tr}\left( \left( \frac{\partial r}{\partial x} \right)^\top G(r) \left( \frac{\partial r}{\partial x} \right) G^{-1}(x) \right) \right.$$

$$+ \sum_{i=1}^{m} \sum_{j=1}^{m} (r^i - x^i)\, g_{ij}\, \mathrm{Tr}\left( \left( \frac{\partial^2 r^j}{\partial x^2} \right) G^{-1} \right)$$

$$\left. + \sum_{k=1}^{m} \sum_{l=1}^{m} \sum_{n=1}^{m} (r^i - x^i) \left( \Gamma_{i;jk} \frac{\partial r^j}{\partial x^l} \frac{\partial r^k}{\partial x^n} g^{ln} - g_{ij} \frac{\partial r^j}{\partial x^k} \Gamma_{ln}^k g^{ln} \right) \right) + \mathcal{O}(\sigma^4),$$

*where $r^i$ is the $i$-th component of $r$, $g_{ij}$ and $g^{ij}$ respectively denote the $(i, j)$ entry of $G$ and $G^{-1}$. The $\Gamma_{k;ij}$ and $\Gamma_{ij}^k$ respectively denote the Christoffel symbol of the first and second kind in $\mathcal{M}$.*

*Proof.* We represent $x \in \mathcal{M}$ and $\tilde{x} \in \mathcal{M}$ in local coordinates as $x \in \mathbb{R}^m$ and $\tilde{x} \in \mathbb{R}^m$, respectively. Furthermore, at $x$, consider a nonlinear function $\phi : \mathbb{R}^m \to \mathbb{R}^m$ that maps the representations in local coordinates of the points near $x$ to those in the normal coordinates; from these settings, $\phi(x) = 0$ holds and let $\tilde{u} = \phi(\tilde{x})$. Denote by $r_u : \mathbb{R}^m \to \mathbb{R}^m$ the reconstruction function represented in the normal coordinates. (The functions $r$ and $r_u$ are then related by $r_u(\tilde{u}) = \phi(r(\tilde{x}))$, and $r_u(0) = \phi(r(x))$ holds.)

Since the distance between $x \in \mathcal{M}$ and $r(\tilde{x}) \in \mathcal{M}$ corresponds to the standard vector norm of the representation of $r(\tilde{x})$ in normal coordinates at $x$, $\mathrm{dist}(r(\tilde{x}), x)^2 = \|r_u(\tilde{u})\|^2$ holds. Near the origin

---

[7]The explicit form of $\eta^i(x, r, r', r'')$ is obtained as

$\eta^i = \sum_{j,k,l,\alpha} g^{ij} \left( \left( \sum_{\beta} -\frac{1}{2} \frac{\partial r^\alpha}{\partial x^k} \frac{\partial g(r)_{\alpha\beta}}{\partial r^j} \frac{\partial r^\beta}{\partial x^l} g^{kl} + \frac{\partial g(r)_{j\alpha}}{\partial r^\beta} \frac{\partial r^\beta}{\partial x^k} \frac{\partial r^\alpha}{\partial x^l} g^{kl} \right) + g(r)_{j\alpha} \left( \frac{\partial^2 r^\alpha}{\partial x^k \partial x^l} g^{kl} \right.$

$\left. + \frac{\partial r^\alpha}{\partial x^l} \left( \frac{\partial g^{kl}}{\partial x^k} + g^{kl} \frac{1}{\rho} \frac{\partial \rho}{\partial x^k} \right) \right) \right)$, where $g^{ij}$ and $g(r)_{ij}$ respectively denote the $(i, j)$ entry of $G^{-1}$ and $G(r(x))$.

of the normal coordinates at x, $r_u^i : \mathbb{R}^m \to \mathbb{R}$ (the $i$-th component of $r_u : \mathbb{R}^m \to \mathbb{R}^m$) admits following Taylor's expansion:

$$r_u^i(\tilde{u}) = r_u^i(0) + \frac{\partial r_u^i}{\partial u} \tilde{u} + \frac{1}{2!} \tilde{u}^\top \left( \frac{\partial^2 r_u^i}{\partial u^2} \right) \tilde{u} + \cdots . \tag{23}$$

Then $\|r_u(\tilde{u})\|^2$ can be expressed as follows:

$$\|r_u(\tilde{u})\|^2 = \sum_{i=1}^m \left( r_u^i(\tilde{u}) \right)^2 \tag{24}$$

$$= \|r_u(0)\|^2 + 2 \, r_u(0)^\top \left( \frac{\partial r_u}{\partial u} \right) \tilde{u} + \tilde{u}^\top \left( \frac{\partial r_u}{\partial u} \right)^\top \left( \frac{\partial r_u}{\partial u} \right) \tilde{u} \tag{25}$$

$$+ \sum_{i=1}^m r_u^i(0) \, \tilde{u}^\top \left( \frac{\partial^2 r_u^i}{\partial u^2} \right) \tilde{u} + \cdots ,$$

where the terms of the order (with respect to $\tilde{u}$) higher than three are omitted in (25). Consider the expectation with respect to $\tilde{u} \sim N(0, \sigma^2 I)$ of (25) (this corresponds to the expectation of $\text{dist}(\text{r}(\tilde{\text{x}}), \text{x})^2$ with respect to $q(\tilde{\text{x}}|\text{x})$). Provided $\sigma^2$ is small, we obtain

$$E_{q(\tilde{\text{x}}|\text{x})} \left[ \text{dist}(\text{r}(\tilde{\text{x}}), \text{x})^2 \right] = E_{q(\tilde{u})} \left[ \|r_u(\tilde{u})\|^2 \right] \tag{26}$$

$$= \|r_u(0)\|^2 + \sigma^2 \, \text{Tr} \left( \left( \frac{\partial r_u}{\partial u} \right)^\top \left( \frac{\partial r_u}{\partial u} \right) \right) + \sigma^2 \sum_{i=1}^m r_u^i(0) \, \text{Tr} \left( \left( \frac{\partial^2 r_u^i}{\partial u^2} \right) \right) + \mathcal{O}(\sigma^4),$$

where $q(\tilde{u})$ denotes the noise density for $\tilde{u} \sim N(0, \sigma^2 I)$, and the terms with an order higher than $\sigma^2$ are assumed to be negligible.

We now represent (26) in local coordinates. For this purpose, the following equations are useful:

$$r_u^i(0) = \phi^i(r(x)) \approx \frac{\partial \phi^i}{\partial x}(r(x) - x), \tag{27}$$

$$\frac{\partial r_u^i}{\partial u^j} = \frac{\partial \phi^i(r(x))}{\partial u^j} = \sum_{k=1}^m \sum_{l=1}^m \frac{\partial \phi^i}{\partial r^k} \frac{\partial r^k}{\partial x^l} \frac{\partial x^l}{\partial u^j}, \tag{28}$$

$$\frac{\partial^2 r_u^i}{\partial u^j \partial u^k} = \left( \sum_{l=1}^m \sum_{n=1}^m \sum_{p=1}^m \sum_{q=1}^m \frac{\partial^2 \phi^i}{\partial r^l \partial r^n} \frac{\partial r^l}{\partial x^p} \frac{\partial r^n}{\partial x^q} \frac{\partial x^p}{\partial u^j} \frac{\partial x^q}{\partial u^k} \right) \tag{29}$$

$$+ \left( \sum_{l=1}^m \sum_{p=1}^m \sum_{q=1}^m \frac{\partial \phi^i}{\partial r^l} \frac{\partial^2 r^l}{\partial x^p \partial x^q} \frac{\partial x^p}{\partial u^j} \frac{\partial x^q}{\partial u^k} \right) + \left( \sum_{l=1}^m \sum_{n=1}^m \frac{\partial \phi^i}{\partial r^l} \frac{\partial r^l}{\partial x^n} \frac{\partial^2 x^n}{\partial u^j \partial u^k} \right) ,$$

$$g_{jk} = \sum_{i=1}^m \frac{\partial \phi^i}{\partial x^j} \frac{\partial \phi^i}{\partial x^k}, \tag{30}$$

$$g^{jk} = \sum_{i=1}^m \frac{\partial x^j}{\partial u^i} \frac{\partial x^k}{\partial u^i}, \tag{31}$$

$$\Gamma_{j;kl} = \sum_{i=1}^m \frac{\partial \phi^i}{\partial x^j} \frac{\partial^2 \phi^i}{\partial x^k \partial x^l}, \tag{32}$$

$$\sum_{j=1}^m \sum_{k=1}^m \Gamma_{jk}^i g^{jk} = - \sum_{j=1}^m \sum_{k=1}^m \frac{\partial^2 x^i}{\partial u^j \partial u^k} \delta^{jk}, \tag{33}$$

where $\frac{\partial x^i}{\partial u^j}$ is the $(i, j)$ entry of $\left( \frac{\partial x}{\partial u} \right) = \left( \frac{\partial \phi}{\partial x} \right)^{-1} \in \mathbb{R}^{m \times m}$, and $\delta^{jk} = 1$ for $j = k$ and $\delta^{jk} = 0$ otherwise in (33). Here, the higher-order terms in (27) can be neglected for small $\sigma^2$, and (28)-(29) are obtained from the chain rule. We can derive (30)-(33) using the properties of the normal coordinates. By applying (27)-(33) to (26) and rearranging the terms, we obtain the result in (22). $\square$

We now provide the first-order necessary conditions for (6) using Lemma 2.

**Lemma 3.** *Provided $\sigma^2$ is small, the derivative of the log of the probability function $\rho_g(x) = \frac{\rho(x)}{\sqrt{\det G(x)}}$ can be approximated for the geometric denoising autoencoder as*

$$\frac{\partial \log \rho_g(x)}{\partial x} = \frac{1}{\rho_g} \frac{\partial \rho_g}{\partial x}(x) = \frac{1}{\rho}\frac{\partial \rho}{\partial x}(x) - \Gamma(x) = G(x)\left(\frac{r(x)-x}{\sigma^2}\right) + \mathcal{O}(\sigma^2), \qquad (34)$$

*where $\Gamma(x) \in \mathbb{R}^m$ is a vector whose $i$-th component is given by $\frac{1}{2}\mathrm{Tr}\left(G^{-1}\frac{\partial G}{\partial x^i}\right)$.*

*Proof.* The first-order necessary conditions for (6) are obtained from the following Euler-Lagrange equations:

$$\frac{\partial \mathcal{L}}{\partial r^i} - \sum_{j=1}^m \frac{\partial}{\partial x^j}\left(\frac{\partial \mathcal{L}}{\partial\left(\frac{\partial r^i}{\partial x^j}\right)}\right) + \sum_{j,k=1}^m \frac{\partial^2}{\partial x^j \partial x^k}\left(\frac{\partial \mathcal{L}}{\partial\left(\frac{\partial^2 r^i}{\partial x^j \partial x^k}\right)}\right) = 0, \qquad i = 1,\ldots,m, \quad (35)$$

where $\mathcal{L}$ denotes (22) approximated according to Lemma 2. Applying $\mathcal{L}$ to (35) results in an equation of the form $r^i - x^i = \sigma^2\,\varphi^i(x,r,r',r'')$ for $i = 1,\ldots,m$, where $\varphi^i$ is a function of $x,r,r',r''$.[8] Assuming $r, G, G^{-1}, \rho$ are smooth and their higher-order derivatives are bounded, we obtain an equation identical to (20) for small $\sigma^2$ by iterating $r^i = x^i + \sigma^2\varphi^i(x,r,r',r'') = x^i + \mathcal{O}(\sigma^2)$. Hence the first-order necessary conditions are obtained as (21). $\qquad\square$

### C.3   Proof of Theorem 1

**Theorem 2.** *Provided $\sigma^2$ is small, the derivative of the log of the probability function $\rho_g(x) = \frac{\rho(x)}{\sqrt{\det G(x)}}$ can be approximated for both the GRCAE and GDAE as*

$$\frac{\partial \log \rho_g(x)}{\partial x} = \frac{1}{\rho_g}\frac{\partial \rho_g}{\partial x}(x) = \frac{1}{\rho}\frac{\partial \rho}{\partial x}(x) - \Gamma(x) = G(x)\left(\frac{r(x)-x}{\sigma^2}\right) + \mathcal{O}(\sigma^2), \qquad (36)$$

*where $\Gamma(x) \in \mathbb{R}^m$ is a vector whose $i$-th component is given by $\frac{1}{2}\mathrm{Tr}\left(G^{-1}\frac{\partial G}{\partial x^i}\right)$.*

*Proof.* Collecting Lemma 1 from Appendix C.1 and Lemma 3 from Appendix C.2 completes the proof. $\qquad\square$

## D   IMPLEMENTATION OF GEOMETRICALLY REGULARIZED AUTOENCODERS

In Appendix D.1, D.2, D.3, and D.4, we provide the implementation details of the geometrically regularized autoencoders for $S^n$, $P(n)$, $N(3)$, and point cloud data considered in our experiments, respectively. We also provide some ideas to deal with the case of manifolds that require multiple coordinate charts in Appendix D.5. We then discuss the case when the data space Riemannian metric is not known *a priori* in Appendix D.6.

### D.1   $S^n$ DATA

#### D.1.1   DATA CORRUPTION PROCESS

Given an $n+1$-dimensional vector with the unit norm $\mathrm{x} \in S^n \subseteq \mathbb{R}^{n+1}$ and a tangent vector $\mathrm{v} \in T_\mathrm{x}S^n \subseteq \mathbb{R}^{n+1}$, the exponential map $\mathrm{Exp}_\mathrm{x} : T_\mathrm{x}S^n \to S^n$ is defined as follows:

$$\mathrm{Exp}_\mathrm{x}(\mathrm{v}) = \cos\|\mathrm{v}\| \cdot \mathrm{x} + \frac{\sin\|\mathrm{v}\|}{\|\mathrm{v}\|} \cdot \mathrm{v}, \qquad (37)$$

---

[8]The explicit form of $\varphi^i(x,r,r',r'')$ is obtained as

$\varphi^i = \sum_{j,k,l,\alpha} g^{ij}\left(-\frac{1}{2}\left(\left(\sum_\beta \frac{\partial g(r)_{\alpha\beta}}{\partial r^j}\frac{\partial r^\alpha}{\partial x^k}\frac{\partial r^\beta}{\partial x^l}g^{kl} + \Gamma_{j;\alpha\beta}\frac{\partial r^\alpha}{\partial x^k}\frac{\partial r^\beta}{\partial x^l}g^{kl} - g_{j\alpha}\frac{\partial r^\alpha}{\partial x^\beta}\Gamma^\beta_{kl}g^{kl}\right)\right.\right.$

$\left. + g_{j\alpha}\frac{\partial^2 r^\alpha}{\partial x^k \partial x^l}g^{kl}\right) + \frac{1}{\rho}\left(\frac{\partial}{\partial x^k}\left(g(r)_{j\alpha}\frac{\partial r^\alpha}{\partial x^l}g^{kl}\rho + \left(\sum_\beta (r^\alpha - x^\alpha)\Gamma_{\alpha;j\beta}\frac{\partial r^\beta}{\partial x^l}g^{kl}\rho - \frac{1}{2}(r^\alpha - x^\alpha)g_{j\alpha}\right.\right.\right.$

$\left.\left.\left.\left.\Gamma^k_{l\beta}g^{l\beta}\rho\right)\right) - \frac{1}{2}\frac{\partial^2}{\partial x^k \partial x^l}\left(g_{j\alpha}(r^\alpha - x^\alpha)g^{kl}\rho\right)\right)\right)$, where $g(r)_{ij}$ is the $(i,j)$ entry of $G(r(x))$.

where $\| \cdot \|$ is the Euclidean norm.

For the data corruption process, we first sample an $n + 1$-dimensional vector $\epsilon \sim N(0, \sigma^2 I)$ and project $\epsilon$ onto $T_x S^n$ (with the origin identified to that of the ambient Euclidean space) as $v = (I - xx^\top)\epsilon$. As discussed in Section 3.2, the input x is then corrupted to $\tilde{x} = \text{Exp}_x(v)$ according to (37).

### D.1.2 THE RECONSTRUCTION ERROR

For a reconstruction function $r : S^n \to S^n$, the reconstruction error is defined as follows:

$$\text{dist}(r(\tilde{x}), x)^2 = \arccos(r(\tilde{x})^\top x)^2. \tag{38}$$

### D.1.3 THE RECONSTRUCTION FUNCTION

We implement our regularized autoencoder for $S^n$ as a mapping $r : S^n \to S^n$. The input and output of the mapping are $n + 1$-dimensional vectors with the unit norm. The reconstruction function r is modeled by a neural network with two hidden layers, with the hyperbolic tangent (Tanh) activation function as follows:

$$r(x) \quad = \quad \text{Proj}(x + W_3 h_2 + b_3), \tag{39}$$
$$h_i \quad = \quad \text{Tanh}(W_i h_{i-1} + b_i), \qquad i = 1, 2, \tag{40}$$

where $h_1, h_2 \in \mathbb{R}^{d_h}$ denote the hidden variables, $h_0$ is set to be $x \in S^n \subseteq \mathbb{R}^{n+1}$, $\text{Proj} : \mathbb{R}^{n+1} \to S^n \subseteq \mathbb{R}^{n+1}$ is the function to project a vector onto a hypersphere by dividing the input vector by its norm, and $W_i, b_i$ for $i = 1, 2, 3$ respectively denote the matrix and vector parameters with sizes defined accordingly as above. We set the hidden variable dimensionality $d_h$ to 1,000 in all the experiments.

### D.1.4 CONTRACTIVE REGULARIZATION

For the case of $S^n$, the contractive term of (5) (without the $\rho(x)$ term) can be computed using r as $\text{Tr}\left(\left(\frac{\partial r}{\partial x}\right)^\top \left(\frac{\partial r}{\partial x}\right)(I - xx^\top)\right)$, where $\frac{\partial r}{\partial x} \in \mathbb{R}^{n+1 \times n+1}$ and $x \in S^n \subseteq \mathbb{R}^{n+1}$.

### D.1.5 AN EMPIRICAL ANALYSIS OF COMPUTATIONAL TIME

Training geometrically regularized autoencoders requires more computations than training vector space autoencoders since it needs to calculate the geodesic distance, exponential map, and geometric contractive term. We have experimentally measured the computational times for the models applying each geometric component explained above to the autoencoder $r : \mathcal{M} \to \mathcal{M}$. We have used the Pytorch library (Paszke et al., 2017) and have utilized NVIDIA Tesla V100 GPU with Intel Xeon E5-2698 v4 2.2 GHz (20-Core) CPU (also for most of the other experiments).

We report in Table 6 the computational times spent for a gradient update iteration for each model with a batch size of 10,000. In the table, AE, GAE, GAE + G., GAE + E., and GAE + C. respectively represent the vanilla autoencoder (of the same input and output dimensions with other models), an autoencoder reflecting the non-Euclidean input and output structure, a model applying the geodesic distance as the reconstruction error for GAE, a model applying the exponential map in data corruption process for GAE, and a model applying the geometric regularization term for GAE. For comparison, we also consider GDAE and GRCAE. To check the time for various data dimensionality, we consider $n = 2, 6, 10, 25, 50, 100$ for $S^n$ data generated as explained in Appendix F.4. From the table, we can observe that applying our geometric components can be scaled to high-dimensional $S^n$ data reasonably well, except for the geometric contractive term, which requires the derivative of the Jacobian $\frac{\partial r}{\partial x} \in \mathbb{R}^{n+1 \times n+1}$ during the computation of gradients for an update.

## D.2 P($n$) DATA

### D.2.1 MATRIX EXPONENTIAL AND MATRIX LOGARITHM

We first provide the closed-form expressions of the matrix exponential on S($n$), the space of $n \times n$ symmetric matrices, and matrix logarithm on P($n$). Given an eigendecomposition of a symmetric

Table 6: The computational times (in milliseconds) spent for a gradient update iteration for autoencoders applying different geometric components for $S^n$ data.

| $n$ | AE | GAE | GAE + G. | GAE + E. | GAE + C. | GDAE | GRCAE |
|-----|------|------|------|------|------|------|------|
| 2 | 5.37 | 5.30 | 7.53 | 6.09 | 35.7 | 10.7 | 39.8 |
| 6 | 5.46 | 5.32 | 7.59 | 6.13 | 68.7 | 10.9 | 74.0 |
| 10 | 5.56 | 5.47 | 7.44 | 6.23 | 103 | 10.9 | 110 |
| 25 | 5.46 | 5.53 | 7.71 | 6.31 | 234 | 11.0 | 248 |
| 50 | 5.68 | 5.58 | 7.79 | 6.45 | 462 | 10.7 | 491 |
| 100 | 6.15 | 6.16 | 8.23 | 7.09 | 1016 | 11.3 | 1100 |

matrix $A = RDR^\top \in S(n)$, with $R \in O(n)$ as the eigenvector matrix and $D = \mathrm{diag}(d_1, \ldots, d_n)$ the matrix of corresponding eigenvalues, the matrix exponential is obtained as

$$\mathrm{Exp}(A) = R \, \mathrm{Exp}(D) \, R^\top, \tag{41}$$

where $\mathrm{Exp}(D) = \mathrm{diag}(\exp(d_1), \ldots, \exp(d_n))$. Given an eigendecomposition of a symmetric positive-definite matrix $A = RDR^\top \in P(n)$ similarly to the above, the matrix logarithm is obtained as

$$\mathrm{Log}(A) = R \, \mathrm{Log}(D) \, R^\top, \tag{42}$$

where $\mathrm{Log}(D) = \mathrm{diag}(\log(d_1), \ldots, \log(d_n))$.

### D.2.2 DATA CORRUPTION PROCESS

Given an input $P \in P(n)$, we first sample a $\frac{n(n+1)}{2}$-dimensional vector $\epsilon \sim N(0, \sigma^2 I)$. As discussed in Section 3.2, the input $P$ is then corrupted to $\tilde{P} \in P(n)$ as follows:

$$\tilde{P} = P^{\frac{1}{2}} \mathrm{Exp}\left([\epsilon]\right) P^{\frac{1}{2}}, \tag{43}$$

where $P^{\frac{1}{2}} = RS^{\frac{1}{2}}R^\top$ for an eigendecomposition of $P = RSR^\top$, and the bracket operator is defined for $\epsilon = (\epsilon_{11}, \ldots, \epsilon_{1n}, \epsilon_{22}, \ldots, \epsilon_{2n}, \ldots, \epsilon_{nn}) \in \mathbb{R}^{\frac{n(n+1)}{2}}$ as $([\epsilon])_{ij} = a_{ij}\epsilon_{ij}$ for $i \leq j$ and $([\epsilon])_{ij} = ([\epsilon])_{ji}$ for $i > j$, with $a_{ij} = 1$ if $i = j$ and $\frac{1}{\sqrt{2}}$ otherwise.

### D.2.3 THE RECONSTRUCTION ERROR

For a reconstruction function $r : P(n) \to P(n)$, the reconstruction error is defined from the affine-invariant Riemannian metric as follows (Fletcher & Joshi, 2007):

$$\mathrm{dist}(r(\tilde{P}), P)^2 = \left\| \mathrm{Log}\left(P^{-\frac{1}{2}} r(\tilde{P}) P^{-\frac{1}{2}}\right) \right\|_F^2, \tag{44}$$

where $P^{-\frac{1}{2}} = RS^{-\frac{1}{2}}R^\top$ for an eigendecomposition of $P = RSR^\top$, $\mathrm{Log} : P(n) \to S(n)$ is the matrix logarithm, and $\|\cdot\|_F$ is the matrix Frobenius norm.

### D.2.4 THE RECONSTRUCTION FUNCTION

The reconstruction functions $r : P(n) \to P(n)$ for GDAE and GRCAE should consider the special structure of $P(n)$. By defining a function $v : P(n) \to S(n)$, the reconstruction function at $P \in P(n)$ is modeled as

$$r(P) = P^{\frac{1}{2}} \mathrm{Exp}(v(P)) P^{\frac{1}{2}}. \tag{45}$$

Hence training $r$ reduces to training unconstrained $v$ (when considering only the lower- or upper-triangular part), and we denote by $v : P(n) \to \mathbb{R}^{\frac{n(n+1)}{2}}$ the function that returns the $\frac{n(n+1)}{2}$-dimensional vector representation of the lower- (or upper-) triangular part of the output for v.

We model $v$ by neural networks with two hidden layers, with the hyperbolic tangent (Tanh) activation function as follows:

$$v(P) = W_3 h_2 + b_3, \tag{46}$$
$$h_i = \mathrm{Tanh}(W_i h_{i-1} + b_i), \qquad i = 1, 2, \tag{47}$$

where $h_1, h_2 \in \mathbb{R}^{d_h}$ denote the hidden variables, $h_0 \in \mathbb{R}^{\frac{n(n+1)}{2}}$ is set to the vector representation of the lower- (or upper-) triangular part of $\mathrm{Log}(P)$ with $\mathrm{Log} : \mathrm{P}(n) \to \mathrm{S}(n)$ as the matrix logarithm,[9] and $W_i, b_i$ for $i = 1, 2, 3$ respectively denote the matrix and vector parameters with sizes defined accordingly as above. We set the dimensionality of the hidden variables $d_h$ to 1,000 for both GDAE and GRCAE in all the experiments.

### D.2.5 Contractive regularization

For the case of $\mathrm{P}(n)$, we can consider the upper- (or lower-) triangular part of the matrix representations as its local coordinates, i.e., $\mathrm{r}(\mathrm{x}) \simeq r(x)$ and $\mathrm{x} \simeq x$, and calculate the contractive term of (5) straight-forwardly.

### D.2.6 An empirical analysis of computational time

Similarly to Appendix D.1.5, we have experimentally measured the computational times for each autoencoder model for $\mathrm{P}(n)$ data. Note that we can calculate the matrix exponential and logarithm faster and numerically more stably using the approximations from Taylor's expansion by assuming small $\sigma^2$ (hence inputs becoming near zero and identity for the matrix exponential and logarithm, respectively, in (43)-(45)). We also precompute and use some quantities frequently appears in (43)-(45) such as $P^{\frac{1}{2}}$ and $P^{-\frac{1}{2}}$.

We consider two different models with different degrees of approximation for GDAE, namely GDAE-v1 and GDAE-v2. GDAE-v1 applies in (43)-(45) the second-order Taylor's expansion on the matrix exponential and logarithm, i.e., $\mathrm{Exp}(\mathrm{v}) \approx I + \mathrm{v} + \frac{1}{2}\mathrm{v}^2$ and $\mathrm{Log}(M) \approx (M - I) - \frac{1}{2}(M - I)^2$. GDAE-v2 additionally applies the first-order Taylor's expansion near $P$ for $\tilde{P}^{\frac{1}{2}}$ and $\mathrm{Log}(\tilde{P})$ required to calculate $\mathrm{r}(\tilde{P})$ based on (45) so that it can avoid performing additional eigenvalue decompositions during training.

We report in Table 7 the computational times for a gradient update iteration for each model with a batch size of 5,000. To check the time for various data dimensionality, we consider $n = 2, 3, 4, 7, 10, 14$ for $\mathrm{P}(n)$ data (with dimensionality $d = \frac{n(n+1)}{2} = 3, 6, 10, 28, 55, 105$) generated as explained in Appendix F.1. From the table, we can observe that applying our geometric components can be scaled to high-dimensional $\mathrm{P}(n)$ data at a rate slower than the linear rate to $d$, except for the geometric contractive term of which the computational time increases at a nearly quadratic rate to $d$ when applied. Also note that even if the GDAE-v2 shows a faster computational time when $n = 2, 3$, it becomes slower than GDAE-v1 as the data dimensionality increases due to the heavy matrix multiplications required in the approximation. We utilize GDAE-v1 for other experiments using $\mathrm{P}(n)$ data.

Table 7: The computational times (in milliseconds) spent for a gradient update iteration for autoencoders applying different geometric components for $\mathrm{P}(n)$ data.

| $n$ | $d$ | AE | GAE | GAE + G. | GAE + E. | GAE + C. | GDAE-v1 | GDAE-v2 | GRCAE |
|---|---|---|---|---|---|---|---|---|---|
| 2 | 3 | 3.02 | 5.10 | 4.66 | 14.6 | 40.1 | 22.1 | 11.2 | 45.0 |
| 3 | 6 | 3.31 | 7.13 | 6.28 | 18.1 | 94.7 | 28.8 | 19.9 | 92.3 |
| 4 | 10 | 3.29 | 8.96 | 8.85 | 25.1 | 213 | 38.1 | 38.8 | 220 |
| 7 | 28 | 3.32 | 19.8 | 19.1 | 40.4 | 1111 | 52.5 | 82.2 | 1126 |
| 10 | 55 | 3.34 | 39.6 | 39.1 | 63.2 | 4573 | 79.2 | 164 | 4594 |
| 14 | 105 | 3.73 | 62.8 | 62.8 | 98.6 | 28949 | 111 | 262 | 28776 |

### D.3 N(3) data

In training GDAE for diffusion tensor imaging (DTI) data in Section 4.4, the structure of the manifold N(3) (using the Fisher information Riemannian metric) should be dealt with properly. Here, we provide technical details for the modeling and training of the reconstruction function $\mathrm{r} : \mathrm{N}(3) \to \mathrm{N}(3)$.

---

[9]It has empirically performed better to use the vector representation obtained from $\mathrm{Log}(P)$ rather than $P$.

### D.3.1 THE FISHER INFORMATION RIEMANNIAN METRIC

Denoting by $N(\mu, \Sigma)$ a normal distribution in $\mathbb{R}^n$ with mean $\mu$ and covariance $\Sigma$, the space of $n$-dimensional normal distributions $\mathrm{N}(n)$ is defined as follows:

$$\mathrm{N}(n) = \{N(\mu, \Sigma) \mid \mu \in \mathbb{R}^n, \Sigma \in \mathrm{P}(n)\}. \tag{48}$$

$\mathrm{N}(n)$ becomes a differentiable manifold with dimension $n + \frac{1}{2}n(n+1)$. By using the Fisher information Riemannian metric with local coordinates $(\mu, \Sigma)$ (for $\Sigma$, only the lower- or upper-triangular is needed part), the inner product of two tangent vectors $V = (V_\mu, V_\Sigma)$, $W = (W_\mu, W_\Sigma)$ for $V_\mu, W_\mu \in \mathbb{R}^n$ and $V_\Sigma, W_\Sigma \in \mathrm{S}(n)$ at a point $N = (\mu, \Sigma) \in \mathrm{N}(n)$ is defined as follows:

$$\langle V, W \rangle_N = V_\mu^\top \Sigma^{-1} W_\mu + \frac{1}{2}\mathrm{Tr}(\Sigma^{-1} V_\Sigma \Sigma^{-1} W_\Sigma). \tag{49}$$

It can be easily verified that the Fisher information metric reduces to the affine-invariant metric when the $\mathbb{R}^n$ part of the tangent vectors is zero; the geodesic distance between two normal distributions with identical means is then obtained in closed-form from the affine-invariant distance between the two covariance matrices. However, in general, the distance between two normal distributions cannot be obtained in closed-form; a numerical algorithm to find the minimal geodesic according to this metric is proposed in Han & Park (2014), along with some physical insights of adopting the Fisher information Riemannian metric for DTI analysis.

### D.3.2 DATA CORRUPTION PROCESS

Given an input voxel data $N = (x, P) \in \mathrm{N}(3)$, we first sample $\epsilon_p \sim N(0, \sigma^2 P)$ and a six-dimensional vector $\epsilon_c \sim N(0, \sigma^2 I)$. The input voxel $(x, P) \in \mathrm{N}(3)$ is then corrupted to $(\tilde{x}, \tilde{P}) \in \mathrm{N}(3)$ as follows:

$$\tilde{x} = x + \epsilon_p, \tag{50}$$

$$\tilde{P} = P^{\frac{1}{2}} \mathrm{Exp}\left([\epsilon_c]\right) P^{\frac{1}{2}}, \tag{51}$$

where $P^{\frac{1}{2}} = R S^{\frac{1}{2}} R^\top$ for an eigendecomposition of $P = R S R^\top$, and the bracket operator $[\,\cdot\,]$ is defined for $\epsilon = (\epsilon_{11}, \epsilon_{12}, \epsilon_{13}, \epsilon_{22}, \epsilon_{23}, \epsilon_{33}) \in \mathbb{R}^6$ as $([\epsilon])_{ij} = a_{ij}\epsilon_{ij}$ for $i \leq j$ and $([\epsilon])_{ij} = ([\epsilon])_{ji}$ for $i > j$, with $a_{ij} = \sqrt{2}$ if $i = j$ and $1$ otherwise. Note that this corresponds to an approximation of the data corruption process $q(\tilde{\mathrm{x}}|\mathrm{x})$ (for small $\sigma^2$) discussed in Section 3.2. In the experiments, we have further applied the first-order Taylor's expansion on the matrix exponential in (51).

### D.3.3 THE RECONSTRUCTION ERROR

Denote the reconstruction function by $\mathrm{r} = (\mathrm{r}_p, \mathrm{r}_c)$, where $\mathrm{r}_p : \mathrm{N}(3) \to \mathbb{R}^3$ corresponds to the location part and $\mathrm{r}_c : \mathrm{N}(3) \to \mathrm{P}(3)$ the value part of the reconstruction function, respectively (we discuss their exact modeling in the later section). Assuming $\sigma^2$ is small, the reconstruction error between $(x, P) \in \mathrm{N}(3)$ and $(\mathrm{r}_\mathrm{p}(\tilde{x}, \tilde{P}), \mathrm{r}_\mathrm{c}(\tilde{x}, \tilde{P})) \in \mathrm{N}(3)$ is approximated according to the inner product of the Fisher information metric defined in (49) as follows:

$$\mathrm{dist}\left((\mathrm{r}_\mathrm{p}(\tilde{x}, \tilde{P}), \mathrm{r}_\mathrm{c}(\tilde{x}, \tilde{P})), (x, P)\right)^2 \tag{52}$$

$$\approx (\mathrm{r}_\mathrm{p}(\tilde{x}, \tilde{P}) - x)^\top P^{-1}(\mathrm{r}_\mathrm{p}(\tilde{x}, \tilde{P}) - x) + \frac{1}{2}\mathrm{Tr}\left(\left(P^{-\frac{1}{2}}\mathrm{r}_\mathrm{c}(\tilde{x}, \tilde{P})P^{-\frac{1}{2}} - I\right)^2\right),$$

where $P^{-\frac{1}{2}} = R S^{-\frac{1}{2}} R^\top$ for an eigendecomposition of $P = R S R^\top$.

### D.3.4 THE RECONSTRUCTION FUNCTION

Next, we model the reconstruction function $\mathrm{r} = (\mathrm{r}_p, \mathrm{r}_c)$ on $\mathrm{N}(3)$. By defining a function $\mathrm{v} : \mathrm{N}(3) \to \mathrm{S}(3)$, the voxel value part $\mathrm{r}_\mathrm{c}$ of the reconstruction function at $(x, P) \in \mathrm{N}(3)$ is modeled as

$$\mathrm{r}_c(x, P) = P^{\frac{1}{2}}\mathrm{Exp}\left(\mathrm{v}(x, P)\right) P^{\frac{1}{2}}. \tag{53}$$

In this way, training $\mathrm{r}_c$ reduces to training unconstrained $\mathrm{v}$ (when considering only the lower- or upper-triangular part), and we denote by $v : \mathrm{N}(3) \to \mathbb{R}^6$ the function that returns the six-dimensional vector representation of the lower- (or upper-) triangular part of the output for $\mathrm{v}$.

We model $\mathrm{r}_p$ and $v$ by neural networks with two hidden layers, with the hyperbolic tangent (Tanh) activation function as follows:

$$
\begin{aligned}
\mathrm{r_p}(x, P) &= x + W_3 h_2 + b_3, & (54) \\
v(x, P) &= W_4 h_2 + b_4, & (55) \\
h_i &= \mathrm{Tanh}(W_i h_{i-1} + b_i), \qquad i = 1, 2, & (56)
\end{aligned}
$$

where $h_1, h_2 \in \mathbb{R}^{d_h}$ denote the hidden variables, and $h_0$ is set to be $(x, p) \in \mathbb{R}^9$ with $p \in \mathbb{R}^6$ as the vector representation of the lower- (or upper-) triangular part of $P$. Moreover, $W_i$, $b_i$ for $i = 1, 2, 3, 4$ respectively denote the matrix and vector parameters with sizes defined accordingly as above. We set the dimensionality of the hidden variables $d_h$ to 1,000.

## D.4 POINT CLOUD DATA

### D.4.1 THE FISHER INFORMATION RIEMANNIAN METRIC FOR POINT CLOUD DATA

A statistical manifold framework has been suggested to deal with point cloud data in Lee et al. (2022). Briefly speaking, they deem each point in a point cloud as a sample from a specific parametric probability density model and identify the data to the density. The Fisher information metric for these density models can then serve as a natural Riemannian metric in the point cloud data space.

For a point cloud data in $\mathbb{R}^D$ represented as $X = \{x_1, \ldots, x_n \mid x_i \in \mathbb{R}^D\}$, by using $X$ itself as the parameter, following parametric probability density model is considered:

$$
p(x; X) = \frac{1}{n\sqrt{|\Sigma|}} \sum_{i=1}^{n} K(\Sigma^{-\frac{1}{2}}(x - x_i)), \tag{57}
$$

where $K : \mathbb{R}^D \to \mathbb{R}$ is a kernel function satisfying $\int_{\mathbb{R}^D} K(u) du = 1$, and $\Sigma \in \mathbb{R}^{D \times D}$ is a symmetric positive-definite bandwidth matrix. In the experiments, we choose the Gaussian kernel function $K(u) = 1/\sqrt{(2\pi)^D} \exp(-u^\top u / 2)$ with $\Sigma = \sigma_p^2 I$ following Lee et al. (2022).

With such a choice of density model, the Fisher information metric $G \in \mathbb{R}^{nD \times nD}$ is obtained as

$$
G_{ijkl} = \int p(x; X) \frac{\partial \log p(x; X)}{\partial X^{ij}} \frac{\partial \log p(x; X)}{\partial X^{kl}} dx \tag{58}
$$

for $i, k = 1, \ldots, n$ and $j, l = 1, \ldots, D$, where $G_{ijkl}$ is the $((i-1)D + j, (k-1)D + l)$ entry of $G$ and $X^{ij}$ is the $j$-th entry of $x_i$. We approximate the integration in (58) as a finite sum of the integrands (with $p(x; X)$ excluded) over the data points $x_1, \ldots, x_n$. We refer the reader to Lee et al. (2022) for more details.

### D.4.2 DATA CORRUPTION PROCESS

Since applying the exponential map based on the Riemannian metric in (58) is computationally very expensive, we resort to an approximation for the data corruption process. Given a vector representation of the point cloud data $\mathrm{x} = (x_1^\top, \ldots, x_n^\top) \in \mathbb{R}^{nD}$, we sample a random vector of the same size $\epsilon \sim N(0, \sigma^2 I)$, and corrupt the data as $\tilde{\mathrm{x}} \approx \mathrm{x} + \sqrt{G^{-1}}\epsilon$, where $G^{-1} \in \mathbb{R}^{nD \times nD}$ is the inverse of the Riemannian metric $G$ in (58). We further approximate $G$ to be a diagonal matrix by considering only the diagonal elements of (58) so that $\sqrt{G^{-1}}$ is computationally tractable.

### D.4.3 THE RECONSTRUCTION ERROR

For a reconstruction function $\mathrm{r} : \mathbb{R}^{n \times D} \to \mathbb{R}^{n \times D}$, consider a point cloud $X = \{x_1, \ldots, x_n \mid x_i \in \mathbb{R}^D\}$ and its reconstructed version $\mathrm{r}(\tilde{X}) = Y = \{y_1, \ldots, y_n \mid y_i \in \mathbb{R}^D\}$. We define the reconstruction error by slightly modifying the Chamfer distance in Yang et al. (2018) as follows (Lee et al., 2022):

$$
\text{Reconstruction error} = \frac{1}{|X|} \sum_{x \in X} \min_{y \in Y} \|x - y\|^2 + \frac{1}{|Y|} \sum_{y \in Y} \min_{x \in X} \|x - y\|^2, \tag{59}
$$

where $|X|$ denotes the number of elements in the set $X$.

Note that using the reconstruction error in (59) does not perfectly correspond to our original definition of GDAE in (6), which uses the geodesic distance as the reconstruction error. However, with a slight abuse of notation, we still use the term GDAE for the method minimizing the reconstruction error defined in (59).

### D.4.4 THE RECONSTRUCTION FUNCTION

We use exactly the same point cloud autoencoder with the reconstruction function $r : \mathbb{R}^{n \times D} \to \mathbb{R}^{n \times D}$ (i.e., the composition of the encoder $f : \mathbb{R}^{n \times D} \to \mathbb{R}^d$ and the decoder $g : \mathbb{R}^d \to \mathbb{R}^{n \times D}$) adopted from the FcNet in Yang et al. (2018). We set the latent space dimensionality $d$ as 512.

### D.5 A REMARK ON IMPLEMENTATION FOR MANIFOLDS THAT REQUIRE MULTIPLE COORDINATE CHARTS

Since the manifolds mainly considered in the experiments, e.g., $S^n$, $P(n)$, $N(n)$, and the statistical manifold for point cloud data, could be embedded in Euclidean spaces, we have directly used the manifold elements embedded in Euclidean spaces as the form of input and output for $r : \mathcal{M} \to \mathcal{M}$.

If this is not the case, e.g., for the abstract manifolds, we can implement the mappings $r : \mathbb{R}^m \to \mathbb{R}^m$ represented in local coordinates. When multiple charts $\{(U_1, x_1), \ldots, (U_C, x_C)\}$ are required,[10] one available approach would be to implement mappings for each chart as separate neural networks and train each mapping by minimizing objective functions weighted by appropriate weight functions. In the case of GDAE, for example, we can solve $C$ optimization problems as follows:

$$\min_{r_i} \int_{\mathcal{M}} E_{q(\tilde{\mathrm{x}}|\mathrm{x})}[\mathrm{dist}(\mathrm{r}_i(\tilde{\mathrm{x}}), \mathrm{x})^2] \rho(x) f_i(x) dx, \quad i = 1, \ldots, C, \tag{60}$$

where $C$ is the number of charts, $r_i : \mathbb{R}^m \to \mathbb{R}^m$ is the mapping for the $i$-th chart (this is $\mathrm{r}_i : U_i \to \mathcal{M}$ represented in local coordinates), and $f_i : \mathcal{M} \to \mathbb{R}_{\geq 0}$ is the weight function for the $i$-th chart satisfying $f_i(x) = 0$ for $\mathrm{x} \notin U_i$, e.g., determined based on partitions of unity (satisfying that, for all $\mathrm{x} \in \mathcal{M}$, there is a neighborhood of $\mathrm{x}$ where all but a finite number of $f_i(x)$s are zero, and $\sum_{i=1}^{C} f_i(x) = 1$). The final reconstruction results for $\mathrm{x} \in \mathcal{M}$ can be obtained by geometrically averaging (after appropriate coordinate transformations) the outputs $r_i(x)$ from each mapping with corresponding weights $f_i(x)$.

Note that there is no guarantee that $r_i(x_i(U_i)) \subseteq x_i(U_i)$ (or $\mathrm{r}_i(U_i) \subseteq U_i$) would always hold. That is, $\mathrm{r}_i(\mathrm{x})$ may be outside of $U_i$ hence $\mathrm{r}_i(\mathrm{x})$ and $\mathrm{x}$ may not be in the same coordinate chart for some $\mathrm{x} \in U_i$. For small $\sigma^2$, such a phenomenon (if it happens) would mostly occur at $\mathrm{x} \in U_i$ near the boundary of $U_i$ since $r_i$ is trained so that $r_i(x)$ is close enough to $x$, i.e., $r_i(x) - x = \mathcal{O}(\sigma^2)$, according to Theorem 1. We can eliminate any undesirable effect that this phenomenon may have on the optimization of the objective function in (60) or on the weighted average of the final results by choosing a weight function $f_i$ that has the value of zero in a sufficiently wide region within $U_i$ that includes the boundary of $U_i$. As a side effect, this choice would also assign relatively large weights for the inner regions of $U_i$ far from the boundary of $U_i$ hence inducing a larger regularization effect (e.g., denoising or contraction) in those regions. Therefore, this may lead to learning the reconstruction function to direct toward the inner regions of $U_i$, which helps to prevent the range of $r_i$ (or $\mathrm{r}_i$) from going outside of $x_i(U_i)$ (or $U_i$).

### D.6 A REMARK ON THE CASE WHEN THE DATA SPACE RIEMANNIAN METRIC IS NOT KNOWN *a priori*

When the Riemannian metric for data space $\mathbb{R}^D$ is not known *a priori*, we can construct the Riemannian metric by resorting to metric learning techniques that either utilize some supervision on the desired distance for a given task or reflect some intuition about the data manifold. For example, in Hauberg et al. (2012), given a set of centers $\{c_1, \ldots, c_K\}$, $c_k \in \mathbb{R}^D$, $k = 1, \ldots, K$ and a set of metrics $\{G_1, \ldots, G_K\}$, $G_k \in \mathbb{R}^{D \times D}$, $k = 1, \ldots, K$ corresponding to each center, a smoothly

---

[10]Here $U_i$ is an open subset of $\mathcal{M}$, and $x_i$ is a homeomorphism of $U_i$ to an open subset of $\mathbb{R}^m$ for $i = 1, \ldots, C$. The collection $\{U_1, \ldots, U_C\}$ covers $\mathcal{M}$, and for all $i$ and $j$, the transition map $x_i \circ x_j^{-1}$ is smooth.

varying Riemannian metric $G : \mathbb{R}^D \to \mathbb{R}^{D \times D}$ is constructed by interpolating the metrics as follows:

$$G(x) = \sum_{k=1}^{K} w_k(x) G_k, \tag{61}$$

where $w_k(x) = \frac{\tilde{w}_k(x)}{\sum_{i=1}^{K} \tilde{w}_i(x)}$, $\tilde{w}_k(x) = \exp(-\frac{1}{2h}(x - c_k)^\top G_k(x - c_k))$, and $h > 0$ is a bandwidth parameter. Here we can obtain the set of metrics $\{G_1, \ldots, G_K\}$ from different metric learning methods, which learn task-specific distance metrics in a supervised manner (Kulis et al., 2013).

In the unsupervised case, based on the intuition that the shortest path in data space should be near the data manifold, not the region where data are sparse, the Riemannian metric $G : \mathbb{R}^D \to \mathbb{R}^{D \times D}$ can be constructed as follows:

$$G(x) = (\alpha \cdot h(x) + \epsilon)^{-1} \cdot I, \tag{62}$$

where $h : \mathbb{R}^D \to \mathbb{R}_{>0}$, $h(x) \to 1$ when $x$ is near the data manifold and $h(x) \to 0$ otherwise, and $\alpha, \epsilon > 0$ are scalars to control the lower and upper bound of the metric, respectively. (Some methods to construct smoothly varying $h(x)$ are provided in Arvanitidis et al. (2021).)

Consider formulating geometrically regularized autoencoders that reflect the Riemannian metrics constructed above. The calculations of the geodesics and exponential maps in these cases usually require numerical solvers (Hauberg et al., 2012). Therefore, efficient training of these autoencoders would require some methods to approximately apply the geometric components, a detailed discussion of which is left for future work.

## E    DETAILS FOR EXPERIMENTS IN THE INTRODUCTION

For the example provided in the introduction, we train the RCAE and the GRCAE (modeled as described in Appendix D.1) on spherical data sampled from the von Mises-Fisher (vMF) distribution with the concentration parameter of 10. To see if the trained results depend on the choice of coordinates, we consider five different spherical coordinate representations of the data for RCAE. We obtain these representations by representing the data with respect to different choices of ambient space reference frames shown in Figure 9 (a) and converting them to spherical coordinate representations. We then train the RCAEs using each representation and compare in the ambient space the reconstruction directions obtained from each RCAE.[11] For the GRCAEs using the ambient space representations, we train the models using data represented in five different reference frames (the same as those considered for RCAE) and compare the reconstruction directions from each model after transforming them to an identical reference frame.

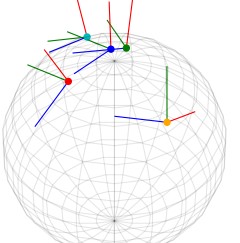 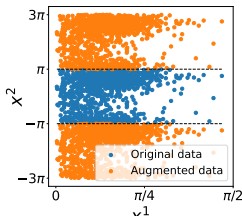

(a) Reference frames considered in the experiments

(b) An example of original and augmented data in spherical coordinates

Figure 9: Additional figures for experiments in the introduction. In (a), red, green, and blue lines represent each reference frame's X-, Y-, and Z-axis, respectively. Note that the reference frames are translated to corresponding spherical coordinate origins for better visualization.

In Figure 1 (b)-(c), we plot the reconstruction directions for a subsample of the data in Figure 1 (a) and provide a magnified view near the zenith for better visualization.

---

[11]To alleviate any topological issues arising from using spherical coordinate representations $(x^1, x^2)$ which parametrizes the three-dimensional points as $(\cos(x^1), \sin(x^1)\cos(x^2), \sin(x^1)\sin(x^2))$, e.g., the fact that, for points with the $x^2$ value near $\pi$ or $-\pi$, nearby points on the sphere can be mapped from two distant regions in spherical coordinates, we augment data obtained by adding or subtracting $2\pi$ from the $x^2$ values (as shown in Figure 9 (b)) during training and only use the original data when getting the reconstruction directions.

# F    DETAILS FOR SECTION 4.1

## F.1    SYNTHETIC DATA GENERATION

We use 10,000 data sampled from $m$ mixtures of tangent space Gaussian on $\mathrm{P}(n)$ in Section 4.1. For the $i$-th mixture, we set the mean as $\mu_i = \mathrm{Exp}(\frac{\sqrt{d}}{2}\mathrm{diag}(e_i)) \in \mathrm{P}(n)$, where $d = \frac{n(n+1)}{2}$ is the dimension of $\mathrm{P}(n)$, $e_i \in \mathbb{R}^n$ is a standard vector whose $i$-th element is one, and $\mathrm{Exp}: \mathrm{S}(n) \to \mathrm{P}(n)$ is the matrix exponential. The covariance of the $i$-th mixture is represented using an orthonormal basis of $T_{\mu_i}\mathrm{P}(n)$ as $\Sigma_i = \mathrm{diag}(\sigma_1, \ldots, \sigma_d) \in \mathbb{R}^{d \times d}$, where $\sigma_k = 0.1$ if $k = 1 + (i-1)n - \frac{(i-1)(i-2)}{2}$ (for $i = 1, \ldots, n$) and $\sigma_k = 0.01$ otherwise, and the $k$-th basis of $T_{\mu_i}\mathrm{P}(n)$ is defined using the bracket operator in (43) as $\mu_i [e_k] \mu_i \in \mathrm{S}(n)$ with $e_k \in \mathbb{R}^d$ as a standard vector whose $k$-th element is one.

## F.2    DEFINITION OF THE ESTIMATION ERROR

For a given data set $\{x_1, \ldots, x_N\}$ represented in local coordinates of $\mathrm{P}(n)$, the log-density gradient estimation error (Est. error) is evaluated as follows:

$$\text{Est. error} = \frac{1}{N}\sum_{i=1}^{N}\left(\frac{\partial \log \rho_g}{\partial x}\Big|_{\text{est}}(x_i) - \frac{\partial \log \rho_g}{\partial x}(x_i)\right)^{\top} G^{-1}(x_i)\left(\frac{\partial \log \rho_g}{\partial x}\Big|_{\text{est}}(x_i) - \frac{\partial \log \rho_g}{\partial x}(x_i)\right), \quad (63)$$

where $\frac{\partial \log \rho_g}{\partial x}\big|_{\text{est}}(x_i)$ and $\frac{\partial \log \rho_g}{\partial x}(x_i)$ respectively denote the estimated value and the ground truth value for $\frac{\partial \log \rho_g(x)}{\partial x}$ at $x_i$. Note that it is usually impossible to calculate the estimation performance based on (63) since the ground truth value of $\frac{\partial \log \rho_g(x)}{\partial x}$ is generally not available. Instead, (63) can be reformulated not to require the $\frac{\partial \log \rho_g(x)}{\partial x}$ term by using the integration by parts technique presented in Hyvärinen (2005). Ignoring the constant terms that do not depend on the estimated value, (63) can be rewritten as

$$\frac{1}{N}\sum_{i=1}^{N}\left(\frac{\partial \log \rho_g}{\partial x}\Big|_{\text{est}}^{\top}(x_i)G^{-1}(x_i)\left(\frac{\partial \log \rho_g}{\partial x}\Big|_{\text{est}}(x_i) + 2\Gamma(x_i)\right) + 2\sum_{j=1}^{m}\sum_{k=1}^{m}\frac{\partial}{\partial x^k}\left(\frac{\partial \log \rho_g}{\partial x^j}\Big|_{\text{est}}(x_i)g^{jk}(x_i)\right)\right), \quad (64)$$

where $\Gamma(x) \in \mathbb{R}^m$ is a vector whose $i$-th component is given by $\frac{1}{2}\mathrm{Tr}\left(G^{-1}\frac{\partial G}{\partial x^i}\right)$. Note that this formulation turns out to be equivalent to the minimization objective of the R-LSLDG method (Ashizawa et al., 2017). In the case of GDAE and GRCAE, further simplification is available for (64): $\sum_j \sum_k \frac{\partial}{\partial x^k}\left(\frac{\partial \log \rho_g}{\partial x^j}\big|_{\text{est}}g^{jk}\right) = \frac{1}{\sigma^2}\mathrm{Tr}\left(\frac{\partial r}{\partial x} - I\right)$ holds from (7). Equation (64) can be used in hyperparameter tuning to evaluate the performance of the estimation in place of (63).

## F.3    TRAINING DAE, RCAE, GDAE, GRCAE, LSLDG, AND R-LSLDG

We model the reconstruction functions $\mathrm{r}: \mathrm{P}(n) \to \mathrm{P}(n)$ for GDAE and GRCAE as explained in Appendix D.2. The reconstruction functions $r: \mathbb{R}^d \to \mathbb{R}^d$ ($d = \frac{n(n+1)}{2}$ for $\mathrm{P}(n)$) for DAE and RCAE are modeled by a neural network with two hidden layers, with the hyperbolic tangent (Tanh) activation function as follows:

$$r(x) = x + W_3 h_2 + b_3, \quad (65)$$
$$h_i = \mathrm{Tanh}(W_i h_{i-1} + b_i), \qquad i = 1, 2, \quad (66)$$

where $h_1, h_2 \in \mathbb{R}^{d_h}$ denote the hidden variables, $h_0$ is set to be $x \in \mathbb{R}^d$, and $W_i, b_i$ for $i = 1, 2, 3$ respectively denote the matrix and vector parameters with sizes defined accordingly as above. We set the dimensionality of the hidden variables $d_h$ to 1,000 for both DAE and RCAE.

We optimize the parameters $\theta = \{W_1, b_1, W_2, b_2, W_3, b_3\}$ in (46), (47) for (6) and (5) to respectively train GDAE and GRCAE (or those in (65), (66) for (1) and (2) to train DAE and RCAE respectively). We apply the Adam algorithm (Kingma & Ba, 2015) using the Pytorch library (Paszke et al., 2017) and update the parameters for 500,000 iterations. We use the batch size of 1,000 to train RCAE and GRCAE and the batch size of 10,000 to train DAE and GDAE. The learning rate starts at 2.5e-5 and is divided by ten after 250,000 iterations with a weight decay parameter of 1e-12. We have applied the weight initialization scheme of Glorot & Bengio (2010). Dividing the initial $W_1, b_1$ by two helped to improve the estimation error for GDAE and GRCAE.

In discretizing (1), (2), (6), and (5), the expectations with respect to the data-generating probability density $\rho(x)$ are approximated by a finite sum over the $N$ training data points with equal weights $\frac{1}{N}$. Given an input x (or $x$), the expectation with respect to the noise density $q(\tilde{x}|x)$ in (6) for GDAE (or $q(\tilde{x}|x)$ in (1) for DAE) is also approximated by a finite sum over finite samples of $\tilde{x}$ (or $\tilde{x}$) with equal weights.

In training the autoencoders, the noise parameter in the data corruption process or the weighting coefficient for the contractive term ($\sigma^2$) should be chosen carefully. In this experiment, $\sigma$ is selected among $\sigma \in \{0.01, 0.025, 0.05\}$ to reduce the modified estimation error in (64), which does not require the true value of $\frac{\partial \log \rho_g(x)}{\partial x}$, on a randomly selected validation data set of sizes 20,000. Since the estimation results from the autoencoders can vary over iterations due to the stochastic nature of the optimization algorithm, instances with the minimum value of (64) during the training process are taken to be the output for each run of autoencoder training.

In LSLDG, the derivative of the log-probability density, i.e., $\frac{\partial \log \rho(x)}{\partial x}$, is modeled as a weighted sum of some basis functions, with the weights optimized in a least-squares sense. The R-LSLDG method estimates the derivative of the log-probability function, i.e., $\sum_j g^{ij} \frac{\partial \log \rho_g(x)}{\partial x^j}$ for $i = 1, \ldots, m$ in local coordinates, similar to LSLDG but using basis functions defined on the Riemannian manifold. For both LSLDG and R-LSLDG, we perform cross-validation for the hyperparameters $\lambda \in \{10^{-3}, 10^{-2}, 10^{-1.5}, 10^{-1}, 10^{-0.5}, 10^0, 10^{0.5}, 10^1\}$ and $\sigma \in \{10^{-4}, 10^{-3}, 10^{-2.5}, 10^{-2}, 10^{-1.5}, 10^{-1}, 10^{-0.5}, 10^0\}$ as described in Sasaki et al. (2014) and Ashizawa et al. (2017).[12] The number of the basis functions is set to 500.

The time spent in training each model is shown in Table 8. The autoencoders (DAE, RCAE, GDAE, and GRCAE) take a much longer time for training when compared to LSLDG and R-LSLDG, which are linear-in-parameter models and have closed-form solutions for the parameters. Furthermore, GRCAE (or RCAE) requires more computations than GDAE (or DAE) due to the contractive terms containing the derivative of the reconstruction functions. Compared to DAE and RCAE, GDAE and GRCAE involve heavier computations such as the eigenvalue decomposition (EVD) to reflect the non-Euclidean geometry of the data. Efficient parallel computation for the eigenvalue decomposition has been available by using the `torch_batch_svd` repository (MIT License).[13]

Table 8: The time (in seconds) spent in training each model for P($n$) data.

| $n$ | LSLDG | DAE | RCAE | R-LSLDG | GDAE | GRCAE |
|---|---|---|---|---|---|---|
| 2 | 9 | 3062 | 17382 | 126 | 20671 | 43254 |
| 3 | 17 | 4087 | 19622 | 306 | 22499 | 67580 |
| 4 | 33 | 4014 | 25120 | 545 | 33166 | 116255 |

### F.4 GEOMETRIC SCORE ESTIMATION OF SYNTHETIC HYPERSPHERICAL DATA

This section provides the result of the geometric score estimation for the data on hypersphere $\mathrm{S}^n = \{p \in \mathbb{R}^{n+1} \mid \|p\| = 1\}$ endowed with the Riemannian metric induced from the Euclidean metric on $\mathbb{R}^{n+1}$. We use 10,000 data generated by mapping samples from $m$ Gaussian mixtures in $T_p\mathrm{S}^n$ via the exponential map $\mathrm{Exp}_p : T_p\mathrm{S}^n \to \mathrm{S}^n$, where $p = (1, 0, \ldots, 0) \in \mathbb{R}^{n+1}$. For the $i$-th mixture, we set the mean as $\mu_i = \frac{1}{\sqrt{2}} e_i \in \mathbb{R}^n$ with $e_i \in \mathbb{R}^n$ as a standard vector whose $i$-th element is one, and the covariance as $\Sigma_i = (0.1)^2 I \in \mathbb{R}^{n \times n}$ so that the ground truth log-probability derivative values are obtainable.

We train the DAE, RCAE, GDAE, GRCAE, LSLDG, and R-LSLDG using the data. For the autoencoder models described in Appendix D.1 (for GDAE and GRCAE) and Appendix F.3 (for DAE and RCAE), we use the Adam optimizer (Kingma & Ba, 2015) of the Pytorch library (Paszke et al., 2017) to update the parameters for 500,000 iterations. We use the batch size of 1,000 to train RCAE and GRCAE and the batch size of 10,000 to train DAE and GDAE. From the initialization scheme of Glorot & Bengio (2010), the initial $W_1, b_1$ are further divided by 1-5 for the experiments. Other optimization and hyperparameter tuning conditions are the same as those explained in Appendix F.3.

---

[12]The definitions of $\lambda$ and $\sigma$ can be found in Sasaki et al. (2014) and Ashizawa et al. (2017).

[13]URL: https://github.com/KinglittleQ/torch-batch-svd.

For both LSLDG and R-LSLDG, we perform cross-validation for the hyperparameters $\lambda \in \{10^{-3}, 10^{-2}, 10^{-1.5}, 10^{-1}\}$ and $\sigma \in \{10^{-2}, 10^{-1.5}, 10^{-1}, 10^{-0.5}, 10^{0}\}$ as described in Sasaki et al. (2014) and Ashizawa et al. (2017). The number of the basis functions is set to 500. The time spent in training each model is shown in Table 9.

Table 9: The time (in seconds) spent in training each model for $S^n$ data.

| $n$ | LSLDG | DAE | RCAE | R-LSLDG | GDAE | GRCAE |
|---|---|---|---|---|---|---|
| 2 | 33 | 3002 | 17165 | 240 | 5236 | 21804 |
| 6 | 111 | 3472 | 20369 | 399 | 6590 | 25864 |
| 10 | 203 | 4112 | 25019 | 570 | 7696 | 34365 |

The estimation errors for $\frac{\partial \log \rho_g}{\partial x}$ obtained from each model are reported in Table 10. Similar to the results in Section 4.1, the GDAE and GRCAE perform much better than other methods (especially for higher dimensionality), while GRCAE gives the least estimation error in most cases.

Table 10: The estimation errors for $\frac{\partial \log \rho_g}{\partial x}$ for tangent space Gaussian mixture data on $S^n$. Bolds represent the best and comparable methods from the t-test with a significance level of 5%.

| $n$ | $m$ | LSLDG | DAE | RCAE | R-LSLDG | GDAE | GRCAE |
|---|---|---|---|---|---|---|---|
| 2 | 2 | **1.44** (0.07) | 4.94 (0.25) | 6.43 (2.05) | 1.77 (0.09) | 1.91 (0.18) | 1.74 (0.15) |
| 6 | 2 | 644 (10.3) | 647 (266) | 586 (93.9) | 5.83 (0.11) | 5.16 (0.16) | **4.74** (0.09) |
| 6 | 6 | 677 (21.1) | 1194 (369) | 1175 (259) | 22.9 (0.14) | **14.2** (1.05) | **14.1** (1.27) |
| 10 | 2 | 782 (6.45) | 630 (107) | 802 (187) | 14.4 (0.35) | 8.52 (0.17) | **7.58** (0.07) |
| 10 | 6 | 1201 (4.46) | 739 (81.1) | 1142 (67.4) | 42.7 (0.12) | **21.3** (3.90) | 20.7 (3.57) |
| 10 | 10 | 1700 (133) | 1380 (54.8) | 1655 (82.2) | 50.4 (0.27) | **39.3** (0.95) | 38.2 (1.72) |

## G DETAILS FOR SECTION 4.2

### G.1 THE RIEMANNIAN LANGEVIN MONTE CARLO (RLMC) METHOD FOR $S^n$

To apply the RLMC method for $S^n$, we should appropriately discretize (9). Recall that, from (9), the Brownian motion $dx_{\mathcal{B}}$ on an $n$-dimensional manifold is written as

$$dx_{\mathcal{B}} = \frac{1}{2}\left(G^{-1}(x)\frac{\partial \log \sqrt{\det G(x)}}{\partial x} + \Psi(x)\right)dt + \sqrt{G^{-1}(x)}dw, \qquad (67)$$

where $dw \in \mathbb{R}^n$ is the Brownian motion in an $n$-dimensional vector space and $\Psi(x) \in \mathbb{R}^n$ is a vector whose $i$-th component is given by $\sum_{j=1}^{n} \frac{\partial g^{ij}}{\partial x^j}$ with $g^{ij}$ as the $(i,j)$ element of $G^{-1}(x) \in \mathbb{R}^{n \times n}$.

Consider spherical coordinate representations $(x^1, \ldots, x^n)$ which parametrizes the points on $S^n$ as $\mathrm{x}(x^1, \ldots, x^n) = (\mathrm{x}^1, \ldots, \mathrm{x}^{n+1}) \in \mathbb{R}^{n+1}$ with $\mathrm{x}^1 = \cos(x^1)$, $\mathrm{x}^i = \prod_{j=1}^{i-1} \sin(x^j)\cos(x^i)$ for $i = 2, \ldots, n$, and $\mathrm{x}^{n+1} = \prod_{j=1}^{n} \sin(x^j)$. The Riemannian metric is then obtained as $G(x) = \mathrm{diag}(1, g_{22}, \ldots, g_{nn}) \in \mathbb{R}^{n \times n}$ with $g_{ii} = \prod_{j=1}^{i-1} \sin^2(x^j)$ for $i = 2, \ldots, n$.

To formulate the RLMC algorithm for $S^n$, it is useful to represent the Brownian motion in (67) in the ambient space. Applying the Ito rule, we can obtain such a representation as follows:

$$d\mathrm{x}_{\mathcal{B}} = \frac{\partial \mathrm{x}}{\partial x}\left(\frac{1}{2}\left(G^{-1}(x)\frac{\partial \log \sqrt{\det G(x)}}{\partial x} + \Psi(x)\right)dt + \sqrt{G^{-1}(x)}dw\right) + \frac{1}{2}\Xi(x)dt, \quad (68)$$

where $\frac{\partial \mathrm{x}}{\partial x} \in \mathbb{R}^{(n+1)\times n}$ is the Jacobian of the parametrization $\mathrm{x}(x)$ with respect to $x$ and $\Xi(x) \in \mathbb{R}^{n+1}$ is a vector whose $i$-th component is given by $\mathrm{Tr}(\sqrt{G^{-1}}^{\top} \frac{\partial^2 \mathrm{x}^i}{\partial x^2}\sqrt{G^{-1}})$ with $\frac{\partial^2 \mathrm{x}^i}{\partial x^2} \in \mathbb{R}^{n \times n}$ as the Hessian of $\mathrm{x}^i$ with respect to $x$.

After a straightforward calculation, the Brownian motion in (68) reduces to

$$d\mathrm{x}_{\mathcal{B}} = -\frac{n}{2}\mathrm{x}dt + Bdw, \qquad (69)$$

where $B = \frac{\partial \mathbf{x}}{\partial x} \sqrt{G^{-1}(x)} \in \mathbb{R}^{(n+1) \times n}$ is a matrix whose column vectors form an orthonormal basis of $T_{\mathbf{x}} \mathbf{S}^n$.[14] Note here that the drift term $(-\frac{n}{2} \mathbf{x} dt)$ is orthogonal to the tangent space $T_{\mathbf{x}} \mathbf{S}^n$.

Each Langevin diffusion step of the RLMC method for $\mathbf{S}^n$ can then be performed by

$$\mathbf{x}_{j+1} = \mathrm{Exp}_{\mathbf{x}_j} \left( \frac{\Delta t}{2} \cdot \mathbf{s}(\mathbf{x}_j) + \sqrt{\Delta t} \cdot (I - \mathbf{x}_j \mathbf{x}_j^\top) \mathbf{w}_j \right), \tag{70}$$

where $\mathbf{x}_j \in \mathbf{S}^n$ is the point sampled at the $j$-th step, $\mathbf{s}(\mathbf{x}_j) \in T_{\mathbf{x}_j} \mathbf{S}^n$ is the estimated geometric score at $\mathbf{x}_j$, and $\mathbf{w}_j \in \mathbb{R}^{n+1}$ is a random vector sampled from $N(0, I)$ with $(I - \mathbf{x}_j \mathbf{x}_j^\top) \mathbf{w}_j$ implying the vector obtained by projecting $\mathbf{w}_j$ onto $T_{\mathbf{x}_j} \mathbf{S}^n$. Note that (70) can be approximated up to the order of $\Delta t$ to the update of $\mathbf{x}_{j+1} = \mathrm{Proj} \left( \mathbf{x}_j + \Delta t \cdot (\frac{1}{2} \mathbf{s}(\mathbf{x}_j) - \frac{n}{2} \mathbf{x}_j) + \sqrt{\Delta t} \cdot (I - \mathbf{x}_j \mathbf{x}_j^\top) \mathbf{w}_j \right)$, which straightforwardly considers the Brownian motion in the ambient space in (69).

The geometric score in (70) can be estimated for a point $\mathbf{x} \in \mathbf{S}^n$ using the reconstruction function $\mathbf{r} : \mathbf{S}^n \to \mathbf{S}^n$ of the GDAE trained from a given set of training data as

$$\mathbf{s}(\mathbf{x}) = \frac{1}{\sigma^2} \mathrm{Log}_{\mathbf{x}}(\mathbf{r}(\mathbf{x})), \tag{71}$$

where the logarithm map $\mathrm{Log}_{\mathbf{x}} : \mathbf{S}^n \to T_{\mathbf{x}} \mathbf{S}^n$, the inverse of the exponential map in (37), is locally defined for an input $\mathbf{y} \in \mathbf{S}^n$ near $\mathbf{x}$ as follows:

$$\mathrm{Log}_{\mathbf{x}}(\mathbf{y}) = \frac{\arccos(\mathbf{x}^\top \mathbf{y})}{\sqrt{1 - (\mathbf{x}^\top \mathbf{y})^2}} \cdot (I - \mathbf{x} \mathbf{x}^\top) \mathbf{y}. \tag{72}$$

### G.2 EXPERIMENTAL DETAILS

We consider four synthetic data sets on $\mathbf{S}^2$: four blobs, two moons, an s-curve, and circles, which are generated by the following procedure. First, we make two-dimensional data sets by using the python `sklearn.datasets` package with zero noise level, place them on the plane $z = 1$, and project them to the spherical surface $\mathbf{S}^2$. Secondly, we add tangent Gaussian noises with standard deviations of 0.01 for two moons, s-curve, circles, and 0.05 for four blobs, and project those perturbed data to the sphere by Riemannian projections. The number of training data is 800, that of validation data is 200, and that of test data is 1000.

For autoencoder-based models, we use the fully-connected neural network architecture that has 3-512-512-512-512-512-3 layers with ReLU-ReLU-linear-ReLU-ReLU-linear activation functions. For S-Flow that uses the RealNVP model (Dinh et al., 2016), the depth is eight and the length of the hidden feature is 512. For all cases, the learning rate is 1e-3 and the weight decay parameter is 1e-12.

For autoencoder-based models, we search $\sigma \in \{0.01, 0.025, 0.05\}$. We use batch gradient descent, and the maximum training epoch is set to 5000. As a validation loss used to determine the best model during the training, we use the modified score estimation error in (64) for autoencoder-based models and negative log-likelihood for S-Flow. In RLMC sampling of the autoencoder-based models, we need to determine the step size $\Delta t$ in (70). The step size is searched over $\Delta t \in \{0.00001, 0.00005, 0.0001, 0.0005, 0.001\}$. We run the experiment five times with different random seeds. In the result table, the averages and standard errors of the best cases are reported, where for each run the best-model-resulting hyperparameters (i.e., $\sigma$ and step size for autoencoder-based models) are selected by computing the MMD metrics (Gretton et al., 2012) between sampled points and the validation data sets. We compute the MMD metric with the exponential kernel $k(x, y) = \exp(-d_{\mathbf{S}^2}^2(x, y)/(2 * \eta^2))$ where $d_{\mathbf{S}^2}(x, y)$ is the squared geodesic distance between data $x, y \in \mathbf{S}^2$ and the bandwidth parameter $\eta$ is defined as the mean of the 1,2,3,4,5-nearest neighbor geodesic distances for all training data.

---

[14] This can be easily verified by the identity of $B^\top B = \sqrt{G^{-1}}^\top \frac{\partial \mathbf{x}}{\partial x}^\top \frac{\partial \mathbf{x}}{\partial x} \sqrt{G^{-1}} = I$ since $G(x) = \frac{\partial \mathbf{x}}{\partial x}^\top \frac{\partial \mathbf{x}}{\partial x}$.

# H DETAILS FOR SECTION 4.3

## H.1 DEFINITION OF THE CLUSTERING TASKS

We define the clusters by merging relevant topics among 20 groups in the Newsgroup20 data set,[15] resulting in 'religion,' 'computers,' 'politics,' 'automobiles,' 'sports,' and a topic in 'science' (one among 'crypt,' 'electronics,' 'med,' and 'space'). We define four clustering tasks by differing the topic in 'science.'

## H.2 TRAINING DAE, GDAE, LSLDG, AND R-LSLDG

The reconstruction function for GDAE is modeled as explained in Appendix D.1, and DAE is modeled in the same way as explained in Appendix F.3.

We then optimize the parameters $\theta = \{W_1, b_1, W_2, b_2, W_3, b_3\}$ for (6) and (1) to train GDAE and DAE, respectively. We use the Adam optimizer (Kingma & Ba, 2015) of the Pytorch library (Paszke et al., 2017) and update the parameters for 500,000 iterations. The learning rate respectively starts at 2.5e-3 and 1e-3 for GDAE and DAE, and is divided by ten after 250,000 iterations. Other optimization conditions are kept the same as those in Appendix F.3, and the corruption noise standard deviations $\sigma$ used are selected between $\sigma \in \{0.025, 0.05\}$ to reduce the modified estimation error in (64) on the training data set.

For both LSLDG and R-LSLDG, we perform cross-validation for the hyperparameters $\lambda \in \{10^{-3}, 10^{-2}, 10^{-1.5}, 10^{-1}\}$ and $\sigma \in \{10^{-1.2}, 10^{-1}, 10^{-0.5}, 10^0\}$ as described in Sasaki et al. (2014) and Ashizawa et al. (2017). The number of the basis functions is set to 300. The time spent in training each model is shown in Table 11.

Table 11: The time (in seconds) spent in training each model for $S^{49}$ data.

| LSLDG | LSLDG Amb | DAE | DAE Amb | R-LSLDG | GDAE |
|-------|-----------|------|---------|---------|------|
| 288   | 166       | 5006 | 5015    | 1435    | 6108 |

## H.3 CLUSTERING OF COVARIANCE MATRIX DATA

In this section we provide a case study of the clustering of covariance matrix data. Following Jayasumana et al. (2015), we calculate covariance matrices from the features of image data and group the images based on the covariance matrices. For this case study, we consider the ETH-80 data set[16] with 3,240 images from eight categories (Leibe & Schiele, 2003), COIL-20 data set[17] with 1,440 images from twenty categories (Nene et al., 1996a), and COIL-100 data set[18] with 7,200 images from a hundred categories (Nene et al., 1996b). In calculating the covariance matrix for each image, we use the feature (from every pixel of the image) consisting of $\{x, y, I, |I_x|, |I_y|\}$, where $x, y$ are the horizontal and vertical pixel locations, $I$ is the intensity of the pixel, and $I_x$ and $I_y$ are the horizontal and vertical gradients of the intensity, respectively.[19] Hence the data in use are in $P(5)$.

We train DAE, GDAE, LSLDG, and R-LSLDG using the covariance matrix data and perform clustering using the trained models similarly as explained in Section 4.3. For the GDAE and DAE models respectively described in Appendix D.2 and Appendix F.3, we use the Adam optimizer (Kingma & Ba, 2015) of the Pytorch library (Paszke et al., 2017) to update the parameters for 400,000 iterations. The learning rate starts at 1e-4 and is divided by ten after 200,000 iterations with a weight decay parameter of 1e-12.

The corruption noise standard deviations $\sigma$ can significantly affect the clustering performance in DAE and GDAE. To choose $\sigma$, we use a (randomly chosen) half of the data as a validation set to train the DAE and GDAE models and measure the clustering performance. The value that gives the

---

[15]URL: http://qwone.com/ jason/20Newsgroups/.

[16]URL: https://github.com/Kai-Xuan/ETH-80.

[17]URL: https://www.cs.columbia.edu/CAVE/software/softlib/coil-20.php.

[18]URL: https://www.cs.columbia.edu/CAVE/software/softlib/coil-100.php.

[19]Each dimension of the features has been normalized before calculating the covariance matrices.

best clustering performance among $\sigma \in \{0.025, 0.05, 0.1, 0.2\}$ is chosen, and we use the rest half of the data to train the models and report the clustering performance. Other optimization conditions are kept the same as those explained in Appendix F.3.

For both LSLDG and R-LSLDG, we perform cross-validation for the hyperparameters $\lambda \in \{10^{-3}, 10^{-2}, 10^{-1.5}, 10^{-1}\}$ and $\sigma \in \{10^{-2}, 10^{-1.5}, 10^{-1}, 10^{-0.5}, 10^{0}\}$ as described in Sasaki et al. (2014) and Ashizawa et al. (2017). The number of the basis functions is set to 100. The clustering is performed using the identical data used to report the results from DAE and GDAE. The time spent in training each model is shown in Table 12.

Table 12: The time (in seconds) spent in training each model for P(5) data.

| Data set | LSLDG | DAE | R-LSLDG | GDAE |
|----------|-------|------|---------|-------|
| ETH-80 | 20 | 1184 | 63 | 14227 |
| COIL-20 | 15 | 1221 | 50 | 13282 |
| COIL-100 | 23 | 1178 | 99 | 14924 |

The averaged clustering performance for five runs of each method is presented in Table 13. For the ETH-80 data set, R-LSLDG and GDAE show better performance than others on average but without much significance. In the case of the COIL-20 data set, the clustering results from DAE show the best performance. The absence of a significant performance difference between DAE and GDAE in both cases may partially be due to the less peculiarity in the covariance matrices, e.g., having extremely small or large eigenvalues, for the relatively small number of categories. (According to the affine-invariant metric, the geometry of the covariance matrices can become more relevant when the relative eigenvalues of the covariance matrices get bigger or smaller.) When the number of categories gets bigger as in the COIL-100 data set, the results from GDAE show better performance with a large margin compared to DAE. Even though this case study does not show a distinct tendency, clustering using GDAE performs well among the considered methods in terms of the overall adjusted Rand index.

Table 13: The adjusted Rand index for the covariance matrix data on P(5). Bolds represent the best and comparable methods from the t-test with a significance level of 5%.

| Data set | LSLDG | DAE | R-LSLDG | GDAE |
|----------|-------|------|---------|-------|
| ETH-80 | **0.318** (0.039) | **0.348** (0.016) | **0.362** (0.035) | **0.361** (0.022) |
| COIL-20 | 0.170 (0.030) | **0.655** (0.012) | 0.483 (0.030) | 0.616 (0.016) |
| COIL-100 | 1.46e-3 (2.9e-4) | 0.204 (0.009) | 0.314 (0.036) | **0.380** (0.015) |

# I  DETAILS FOR SECTION 4.4

## I.1  TRAINING OF THE RECONSTRUCTION FUNCTION

We model GDAE for N(3) as explained in Appendix D.3.4. We then optimize the parameters $\theta = \{W_1, b_1, W_2, b_2, W_3, b_3, W_4, b_4\}$ for (6) to train GDAE. We model DAE as explained in Appendix F.3 with $d = 9$, and the parameters in (65), (66) are optimized for (1). We use the Adam optimizer (Kingma & Ba, 2015) of the Pytorch library (Paszke et al., 2017) and update the parameters for 40,000 iterations. The learning rate starts at 1e-4 and is divided by 100 after 20,000 iterations. In discretizing (6), the expectation with respect to the data-generating probability density $\rho(x)$ and the noise density $q(\tilde{x}|x)$ are dealt with in the same way as for the case studies in Section 4.1 (explained in Appendix F.3), and we use the approximated reconstruction error in (52). We use the corruption noise standard deviation $\sigma = 0.05$ in training both DAE and GDAE.

## I.2  A GDAE-BASED DTI FILTERING ALGORITHM

Based on the discussions in Section 4.4, we summarize a DTI filtering algorithm based on the GDAE in Algorithm 1. At each iteration, the input points are updated toward the reconstructed points of GDAE along the geodesic with a predefined step size $\epsilon$. Assuming small step sizes, we approximate the update along the geodesic on N(3) by treating the updates of the location (in $\mathbb{R}^3$) and the value (in

P(3)) separately. The update of the voxel values then reduces to update along the geodesic according to the affine-invariant Riemannian metric on P(3) (Fletcher & Joshi, 2007), as expressed in line 5 of Algorithm 1. The functions $\text{Exp}_Q : S(3) \to P(3)$ and $\text{Log}_Q : P(3) \to S(3)$ (for $Q \in P(3)$) are defined as follows:

$$\text{Exp}_Q(A) = Q^{\frac{1}{2}} \text{Exp}\left(Q^{-\frac{1}{2}} A Q^{-\frac{1}{2}}\right) Q^{\frac{1}{2}}, \tag{73}$$

$$\text{Log}_Q(P) = Q^{\frac{1}{2}} \text{Log}\left(Q^{-\frac{1}{2}} P Q^{-\frac{1}{2}}\right) Q^{\frac{1}{2}}, \tag{74}$$

where $Q^{\frac{1}{2}} = RS^{\frac{1}{2}}R^\top$ and $Q^{-\frac{1}{2}} = RS^{-\frac{1}{2}}R^\top$ for an eigendecomposition of $Q = RSR^\top$, and the functions $\text{Exp}(\cdot)$ and $\text{Log}(\cdot)$ respectively denote the matrix exponential and logarithm of which the closed-forms are provided in Appendix D.2.1. When the iteration is terminated, we assign the final voxel values to their original locations.

---

**Algorithm 1** DTI Filtering Algorithm Using GDAE

---

**Given:** Voxels of a DTI datum $(x_i, P_i) \in N(3)$, $i = 1, \ldots, N$.
**Input:** Reconstruction function r : $N(3) \to N(3)$ of the GDAE trained from the given set of voxels, step size $\epsilon$, the number of iterations $N_{\text{iter}}$.
**Initialize:** $(y_{i,1}, Q_{i,1}) = (x_i, P_i)$ for $i = 1, \ldots, N$.
**Iteration:**
1: **for** $j = 1, \ldots, N_{\text{iter}}$, $i = 1, \ldots, N$ **do**
2:     Compute the reconstruction point $(y_i, Q_i) = \text{r}(y_{i,j}, Q_{i,j})$.
3:     Shift the current point according to step size $\epsilon$:
4:         $y_{i,j+1} \leftarrow y_{i,j} + \epsilon(y_i - y_{i,j})$,
5:         $Q_{i,j+1} \leftarrow \text{Exp}_{Q_{i,j}}\left(\epsilon \text{Log}_{Q_{i,j}}(Q_i)\right)$ ($Q_{i,j+1} \leftarrow Q_i$ holds when $\epsilon = 1$).
6: **end for**
7: Assign $P_{i,f} = Q_{i,N_{\text{iter}}+1}$ for $i = 1, \ldots, N$.
**Output:** Filtered voxels $(x_i, P_{i,f}) \in N(3)$, $i = 1, \ldots, N$.

---

## I.3 DETAILS FOR N(2) FILTERING EXPERIMENTS

In this experiment, we consider a synthetic N(2) data set obtained by gathering the position (in $\mathbb{R}^2$)-value (in P(2)) pairs at 1,600 grid points (i.e., $N = 1,600$) of a P(2) field on $\mathbb{R}^2$. For the P(2) field for clean data, we gather four distinct smoothly-varying P(2) fields as depicted in Figure 5. For the noisy data, we corrupt each value in P(2) as explained in (43) with noise standard deviations (or noise levels) in $\{0.02, 0.05, 0.1, 0.2\}$.

We model the neural networks for both DAE and GDAE as explained in Appendix F.3 and Appendix D.3.4 with appropriate modifications to deal with data in N(2), respectively, and train the reconstruction functions. We use the Adam optimizer (Kingma & Ba, 2015) and update the parameters for 20,000 iterations. The learning rate starts at 1e-4 and is divided by 100 after 10,000 iterations. The corruption noise standard deviation is chosen among $\sigma \in \{0.01, 0.025, 0.05, 0.1\}$ to show the best modified estimation error in (64). For the trained models, we apply Algorithm 1 (with appropriate modifications to deal with N(2) data) with $N_{\text{iter}} \leq 300$ and the step size $\epsilon \in \{0.01, 0.03, 0.1, 0.3, 1.0\}$ chosen to achieve the minimum averaged square distances (according to the affine-invariant metric) to the clean data.

For the MVKR method in Banerjee et al. (2015), the number of considered nearest neighbors $k \in \{10, 20, 30, 40, 100\}$ and the kernel bandwidth parameter $\sigma \in \{1, 10, 20, 30, 50\}$ are chosen to achieve the minimum averaged square distances (according to the affine-invariant metric) to the clean data.

We report the R-squared score to compare the filtering results from each method in Table 4, and the scores are calculated as follows:

$$\text{R-squared score} = 1 - \frac{\sum_i \text{dist}(P_{i,f}, P_{i,true})^2}{\sum_i \text{dist}(\bar{P}, P_{i,true})^2}, \tag{75}$$

where $\text{dist}(P_1, P_2)$ is the distance between $P_1, P_2 \in \text{P}(2)$ obtained according to the affine-invariant metric, $P_{i,f} \in \text{P}(2)$ is the $i$-th filtered value, $P_{i,true} \in \text{P}(2)$ is the $i$-th value of the clean data, and $\bar{P} \in \text{P}(2)$ is the intrinsic mean of $\{P_{1,true}, \ldots, P_{N,true}\}$ (Moakher, 2005).

### I.4 PREPROCESSING DIFFUSION TENSOR IMAGING (DTI) DATA

Data used in the preparation of the experiments performed in Section 4.4 were obtained from the Alzheimer's Disease Neuroimaging Initiative (ADNI) database (adni.loni.usc.edu).[20] Among the data set, we use the DTI data for a randomly chosen healthy subject.

FSL library (Jenkinson et al., 2012) is utilized to fit diffusion tensors from raw diffusion-weighted image data.[21] The brain extraction tool (`BET`), eddy current correction function (`eddy_correct`), a linear transformation tool to match the brain template (`flirt`), and `dtifit` module are used in the preprocessing of the DTI data. The region of interest (ROI) is chosen as an axial slice that intersects with the corpus callosum region of the brain.

## J DETAILS FOR SECTION 4.5

Overall experimental settings have been set similarly to those in Lee et al. (2022). We add noise to each point $x \in \mathbb{R}^3$ in the point cloud of the data sets (ShapeNet, ModelNet10, and ModelNet40) according to $x \to x + m \cdot v$, where $v \in \mathbb{R}^3 \subseteq \text{S}^2$ is uniformly sampled on the unit sphere and $m$ is sampled from the Gaussian distribution with zero mean and different levels of standard deviation (0.01, 0.05, 0.1, and 0.2 of the diagonal length of the point cloud bounding box) as done in Lee et al. (2021a).

For the parameters in the kernel functions (57), we use $\Sigma = \sigma_p^2 I$ with $\sigma_p = 0.8 \times \text{MED}$, where MED denotes the median of the distances between the points in the point cloud and their nearest points. The mean values of MEDs of each data set are 0.0320, 0.0364, 0.0442, and 0.579 for the cases of the noise levels 0.01, 0.05, 0.1, and 0.2, respectively.

To train the autoencoders defined in Appendix D.4, we use ADAM (Kingma & Ba, 2015) with a learning rate of 1e-4, betas of [0.9, 0.999], a weight decay of 1e-6, and a batch size of 16; the total number of epochs is 500. The noise parameters $\sigma$ for DAE and GDAE are set to 0.1 and 1e-4, respectively. The coefficient for the regularization term (defined in (13) of Lee et al. (2022)) is set to $\lambda = 8000$ for both the 'AE + R.' and 'GDAE + R.' methods in Table 5.

---

[20]The ADNI was launched in 2003 as a public-private partnership, led by Principal Investigator Michael W. Weiner, MD. The primary goal of ADNI has been to test whether serial magnetic resonance imaging (MRI), positron emission tomography (PET), other biological markers, and clinical and neuropsychological assessment can be combined to measure the progression of mild cognitive impairment (MCI) and early Alzheimer's disease (AD).

[21]The license information is available at URL: https://fsl.fmrib.ox.ac.uk/fsl/fslwiki/Licence.

