# OpenReview forum: "Geometrically regularized autoencoders for non-Euclidean data"
_ICLR.cc/2023/Conference — ICLR 2023 poster_

### Official Review · Reviewer_hFuw · 2022-10-23

**Confidence:** 3
**Correctness:** 3
**Technical Novelty And Significance:** 4
**Empirical Novelty And Significance:** 3
**Recommendation:** 6

**Clarity, Quality, Novelty And Reproducibility:**

Overall the paper is well written. But as suggested before, language can be made more accessible. For example, I think the notation of Dirichlet energy can be introduced right after Eq. (2). Explain its effect as the regularization term in Eq. (2). Then in section 3.1, manifest that this paper generalizes this conventional Dirichlet energy for manifold case.

The paper seems novel to me.

**Strength And Weaknesses:**

The paper has some features that can be considered both as strength and weakness.

1. Insights from differential geometry and manifold learning are introduced. This could educate the machine learning community but also has the risk of not being appreciated.

2. The paper shows extensive empirical results. However, most of them aims at showing a more accurate estimate of geometric score. That is, the gradient of log probability distribution function. While I agree it is necessary to show effectiveness using intrinsic measure like this, I think the community may care more about the practical implications.

Another weakness is that the method relies on the metric $G(x)$ at each local coordinate. $G(x)$ may not be known a priori. In fact, $G(x)$ is often unknown for many learning problems. The paper should discuss how to handle it in a separate section. Will it require more training data, compared against the conventional regularizer, to figure out the $G(x)$?

**Summary Of The Paper:**

The paper proposes a regularization technique for auto-encoders, when the data is non-Euclidean. Conventional regularizers encourage small MSE reconstruction error when input is noised. In contrast, the proposed method replaces MSE reconstruction error with geodesic distances along the data manifold. Empirical study shows that the method is better at estimating geometric score, and could lead to more useful learned features.

**Summary Of The Review:**

The paper imports notions from differential geometry and manifold learning. This could have pros and cons.

---

> ### Author Response · Authors · 2022-11-18
> **Reply to reviewer's comments #1**
>
> ### About weaknesses
> > **[W1]** Insights from differential geometry and manifold learning are introduced. This could educate the machine learning community but also has the risk of not being appreciated.
>
> **[A1]**
> - In **Appendix A** of the revision, we have included some mathematical backgrounds on differentiable manifolds and Riemannian geometry required to formulate our geometric autoencoder regularization.
> - We believe that introducing this background will make our paper more accessible to a broader readership in the machine learning community so that our work can be more appreciated.
>
> > **[W2]** The paper shows extensive empirical results. However, most of them aims at showing a more accurate estimate of geometric score. That is, the gradient of log probability distribution function. While I agree it is necessary to show effectiveness using intrinsic measure like this, I think the community may care more about the practical implications.
>
> **[A2]**
> - The geometric score estimation experiments in **Section 4.1** has been performed to verify **Theorem 1** experimentally and to provide a solid basis for future applications of autoencoders. Subsequent experiments have been conducted targeting the application itself rather than demonstrating the accuracy of geometric score estimation (although it may have the effect of showing that).
> - Since we have provided various application examples such as data sampling (**Section 4.2**), clustering (**Section 4.3**), noise filtering (**Section 4.4**), and robust representation learning (**Section 4.5**) beyond the accurate geometric score estimation, and have shown promising results using real-world non-Euclidean data, we believe that our study provides sufficient practical implications for the machine learning community.
> - To emphasize these aspects, we have revised the first paragraph of **Section 4** of the revision as follows:
> > 'In the experiments, we first demonstrate the geometric score estimation (**Theorem 1**) based on our geometrically regularized autoencoders for non-Euclidean data, providing a solid basis for future applications of autoencoders. We then utilize the proposed autoencoders for various applications, such as data sampling based on the Langevin Monte Carlo methods (Girolami & Calderhead, 2011) or clustering and noise filtering based on mode-seeking (Fukunaga & Hostetler, 1975; Cheng, 1995; Comaniciu & Meer, 2002), involving real-world non-Euclidean data sets. We also examine the usefulness of the proposed autoencoders in the representation learning perspective, using noisy point cloud data.'
>
> > **[W3]** Another weakness is that the method relies on the metric $G(x)$ at each local coordinate. $G(x)$ may not be known a priori. In fact, $G(x)$ is often unknown for many learning problems. The paper should discuss how to handle it in a separate section. Will it require more training data, compared against the conventional regularizer, to figure out the $G(x)$?
>
> **[A3]**
> - When the Riemannian metric for data space $\mathbb{R}^D$ is not known
> *a priori*, we can construct the Riemannian metric $G: \mathbb{R}^D \rightarrow \mathbb{R}^{D\times D}$ by resorting to metric learning techniques that either utilize some supervision on the desired distance for a given task or reflect some intuition about the data manifold.
> We have provided some methods we can consider in constructing the Riemannian metric based on recent metric learning approaches [1,2,3] in **Appendix D.6** of the revision.
> - We can then formulate geometrically regularized autoencoders that reflect the constructed Riemannian metrics.
> Note that when reflecting these Riemannian metrics in our geometrically regularized autoencoders, the calculations of the geodesics and exponential maps usually require numerical solvers [1].
> Therefore, efficient training of these autoencoders would require some methods to approximately apply the geometric components, a detailed discussion of which is left for future work.
> - We have discussed these aspects at the end of **Appendix D.6** of the revision.
> - References:
> > [1] Hauberg, S., Freifeld, O., & Black, M. (2012). A geometric take on metric learning. Advances in Neural Information Processing Systems, 25. \
> [2] Kulis, B. (2013). Metric learning: A survey. Foundations and Trends® in Machine Learning, 5(4), 287-364.\
> [3] Arvanitidis, G., Hauberg, S., & Schölkopf, B. (2021, March). Geometrically Enriched Latent Spaces. In International Conference on Artificial Intelligence and Statistics (pp. 631-639). PMLR.

---

> > ### Comment · Reviewer_hFuw · 2022-12-08
> > **keep my original rating**
> >
> > The authors addressed my concerns in the revised draft. However, that doesn't change the nature of this work. My rating therefore stays at 6.

---

> > > ### Author Response · Authors · 2022-12-10
> > > **Thank you**
> > >
> > > We appreciate the reviewer providing constructive feedback to improve our paper.

---

> ### Author Response · Authors · 2022-12-06
> **Reply to reviewer's comments #2**
>
> ### About clarity, quality, novelty and reproducibility
> > **[C1]** Overall the paper is well written. But as suggested before, language can be made more accessible. For example, I think the notation of Dirichlet energy can be introduced right after Eq. (2). Explain its effect as the regularization term in Eq. (2). Then in section 3.1, manifest that this paper generalizes this conventional Dirichlet energy for manifold case.
>
> **[A-C1]**
> - We have revised the manuscript according to the reviewer's suggestion.

---

### Official Review · Reviewer_F6Np · 2022-10-24

**Confidence:** 3
**Correctness:** 3
**Technical Novelty And Significance:** 3
**Empirical Novelty And Significance:** 2
**Recommendation:** 6

**Clarity, Quality, Novelty And Reproducibility:**

### Clarity:
I found the motivation of the paper to be quite clear, but I struggled with the more mathematical parts of the paper. I would have valued a bit more hand-holding along the way.

### Quality:
The work appears sensible, and I particularly enjoyed the denoising autoencoder extension to manifolds. That's very clean and easy to understand. I have a hard time judging the remaining parts of the paper as I didn't quite understand the math.

### Novelty:
As far as I can tell, the paper is quite novel.

### Reproducibility
I would not be able to reproduce the experiments from the paper alone. I'm happy to see that code will be released, though.



**Strength And Weaknesses:**

## Strengths
* The paper approaches an important problem as regularization on manifolds is generally understudied.
* The idea of extending denoising autoencoders to manifold data is neat as it is really simple to work with. I very much appreciate the practicalities of this idea.
* The empirical study is quite broad as several data sets from different manifolds are considered (see also Weaknesses below).

## Weaknesses
* In general, I found the paper somewhat difficult to read. As such, the weaknesses listed here may very well be weaknesses of the reader rather than of the paper.
* The opening motivation for the paper is that standard vector-based regularizers do not work on manifolds. While I generally believe this to be true, I found the demonstration thereof to be somewhat thin. I appreciate the idea of empirically demonstrating the claim, but I think this demonstration should be more elaborate.
* I didn't understand the sentence (page 4), "use the weighted volume element $\rho(x) dx$ instead of $\sqrt{\det G} dx$." Does this mean that you ignore the volume element of the metric? That sounds dodgy.
* I did not understand Eq. 5, in particular the trace-term. Is my reading correct that $\left(\frac{\partial r}{\partial x}\right)^T G \left(\frac{\partial r}{\partial x}\right)$ is a scalar, which can be moved outside the trace-function? I did not understand why the inverse metric appears in this expression. Perhaps it's obvious, but it wasn't to me.
* I also had trouble understanding Eq 8. Why does this use the inverse metric? I would have intuitively guessed that it should be the metric (not its inverse). Can some explanation be provided?
* The experiments generally focus on very low-dimensional manifolds (e.g. 3x3 symmetric positive definite matrices). I would expect that autoencoder-style models have most value on high-dimensional data. I suppose that the method has difficulty scaling to high-dimensional data; what is the main bottleneck?

## Minor issues
* The paper generally talks of Riemannian latent spaces, but, as far as I could tell, mostly Euclidean latent spaces are considered. Perhaps one should here call on one of the many papers which argue that autoencoder-style latent spaces should be viewed as Riemannian (through the pull-back metric). See e.g. https://arxiv.org/abs/1710.11379

**Summary Of The Paper:**

The paper proposes an autoencoder for data residing on a Riemannian manifold alongside a geometric regularize. The reconstruction trivially replaces Euclidean distances with geodesics ones, and the bulk of the paper is concerned with the regularization. I did not fully understand the regulerization scheme (details below), but it seems related to denoising and contractive autoencoders. Empirically, the method is shown to work well on low-dimensional data; it is unclear if the method will scale to high-dimensional manifolds, which arguably is where the dimensionality reduction associated with autoencoders is most valuable.

**Summary Of The Review:**

A promising paper, that I did not fully understand. If I understand the work better during the rebuttal phase I expect to have a more positive vote.

---

> ### Author Response · Authors · 2022-11-18
> **Reply to reviewer's comments #1**
>
> ### About weaknesses
> > **[W1]** In general, I found the paper somewhat difficult to read. As such, the weaknesses listed here may very well be weaknesses of the reader rather than of the paper.
>
> **[A1]**
> - In **Appendix A** of the revision, we have provided some mathematical backgrounds on differential geometry and Riemannian manifolds required to formulate our geometrically regularized autoencoders, e.g., the coordinate charts, Riemannian metric, geodesics, exponential map, natural volume element, and integration on manifolds.
> - In **Appendix B.1** of the revision, we have also included a short verification of the coordinate transformation rules considered in the paper.
> - We hope introducing these backgrounds can help the reviewers and other readers understand our paper more easily.
>
> > **[W2]** The opening motivation for the paper is that standard vector-based regularizers do not work on manifolds. While I generally believe this to be true, I found the demonstration thereof to be somewhat thin. I appreciate the idea of empirically demonstrating the claim, but I think this demonstration should be more elaborate.
>
> **[A2]**
> - We have refined the example in the introduction to address the reviewer's concern.
> To see if the standard vector space autoencoder regularizations depend on the choice of coordinates, we have trained RCAEs using five different spherical coordinate representations of the training data.
> Similarly, for the GRCAEs using the ambient space representations, we have trained the models using data represented in five different reference frames.
> - From **Figure 1** on page 2 of the revision, we can observe that the trained reconstruction functions of RCAE can heavily depend on the choice of coordinates.
> Moreover, they sometimes fail to learn the correct contractive directions towards regions with dense data, especially near the singularity (or the spherical coordinate origins).
> - On the other hand, GRCAEs can recover those directions successfully and show results almost invariant to the choice of coordinates.
> - We have summarized the details of this experiment in **Appendix E** of the revision.
>
> > **[W3]** I didn't understand the sentence (page 4), "use the weighted volume element $\rho(x)dx$ instead of $\sqrt{\det G}dx$." Does this mean that you ignore the volume element of the metric? That sounds dodgy.
>
> **[A3]**
> - We have used the weighted volume element $\rho(x) \ dx \equiv \rho_g(x) \sqrt{\det G(x)} dx$ in formulating our autoencoders as a straightforward generalization of the vector space autoencoders in Eqs (1) and (2), which use the volume element of the form $\rho(x) \ dx$.
> - Note that $\rho(x) \ dx$ can be a valid volume element on $\mathcal{M}$, which satisfies the coordinate transformation rule of the volume elements.
> (We can easily verify this fact by observing that, under a coordinate transformation $x \mapsto x' = \phi(x)$ (then $G(x) \mapsto G'(x')$, $\rho_{g}(x) \mapsto \rho_{g}'(x')$, and $\rho(x) \mapsto \rho'(x')$), $\sqrt{\det G(x)} dx = \sqrt{\det G' (x')} dx'$ and $\rho_{g}(x) = \rho_{g}'(x')$ hold for the natural volume element and a positive scalar function defined on $\mathcal{M}$, respectively, so $\rho(x) \ dx = \rho'(x') \ dx'$ holds.)
> - To avoid any misleading, we have rephrased the sentence as
> > 'replace the natural volume element $\sqrt{\det G} \ dx$ with the weighted volume element $\rho(x) \ dx$.'

---

> ### Author Response · Authors · 2022-11-18
> **Reply to reviewer's comments #2**
>
> ### About weaknesses
> > **[W4]** I did not understand Eq. 5, in particular the trace-term. Is my reading correct that
>  $\left(\frac{\partial r}{\partial x}\right)^T G \left(\frac{\partial r}{\partial x}\right)$
>  is a scalar, which can be moved outside the trace-function? I did not understand why the inverse metric appears in this expression. Perhaps it's obvious, but it wasn't to me.
>
> **[A4]**
> - Since $\frac{\partial r}{\partial x}$ is an $m \times m$ matrix, $\left(\frac{\partial r}{\partial x}\right)^\top G(r) \left(\frac{\partial r}{\partial x}\right)$ is also an $m\times m$ matrix.
> - We need the $G^{-1}$ term to make the objective function in Eq (5) be invariant to the choice of local coordinates, i.e., differential geometrically speaking, be an intrinsic quantity.
> - The coordinate-invariance of Eq (5) can be verified by replacing $f$ with $r$ and $H(f(x))$ with $G(r(x))$ in the discussion in **Appendix B.1** of the revision.
> - That is, under a local coordinate transformation $x \mapsto x’ = \phi(x)$ (then $r(x) \mapsto r’(x’) = \phi(r(x))$ holds), \
> (i) $G(x) \mapsto G’(x’) = \Phi^{-\top} G(x) \Phi^{-1}$, where $\Phi = \left. \frac{ \partial \phi }{ \partial x }\right||_ x$, \
> (ii) $G(r(x)) \mapsto G’(r’(x’)) = \Psi^{-\top} G(r(x)) \Psi^{-1}$, where $\Psi = \left.\frac{\partial \phi}{\partial x}\right||_{r(x)}$, and \
> (iii) $\frac{\partial r}{\partial x} \mapsto \frac{\partial r’}{\partial x’} = \Psi \frac{\partial r}{\partial x} \Phi^{-1}$ \
> hold hence we can verify the coordinate invariance, i.e., $\text{Tr}\left( \left( \frac{\partial r}{\partial x} \right)^\top G(r) \left( \frac{\partial r}{\partial x} \right) G^{-1} \right) = \text{Tr}\left( \left( \frac{\partial r'}{\partial x'} \right)^\top G'(r') \left( \frac{\partial r'}{\partial x'} \right) G'^{-1} \right)$.
> - Without the $G^{-1}$ term, the coordinate invariance cannot be generally satisfied.
>
> > **[W5]** I also had trouble understanding Eq 8. Why does this use the inverse metric? I would have intuitively guessed that it should be the metric (not its inverse). Can some explanation be provided?
>
> **[A5]**
> - Similar to the discussion in **[A4]** above, this estimation error is also defined to be coordinate-invariant.
> - Under a coordinate transformation $x \mapsto x’ = \phi(x)$, according to the chain rule, $\frac{\partial \log \rho_g}{\partial x} \mapsto \frac{\partial  \log \rho_g}{\partial x’} = \Phi^{-\top} \frac{\partial \log \rho_g}{\partial x} $, where $\Phi = \frac{\partial \phi}{\partial x}$ (here $\frac{\partial \log \rho_g}{\partial x} \in \mathbb{R}^m$ is written as a column vector).
> - According to the coordinate transformation rule for the Riemannian metric used in **Appendix B.1** of the revision, $G^{-1} \mapsto {G'}^{-1} = \Phi G^{-1} \Phi^\top$ holds.
> - Therefore we can verify that $\frac{\partial \log \rho_g}{\partial x}^\top G^{-1} \frac{\partial \log \rho_g}{\partial x} = \frac{\partial \log \rho_g}{\partial x’}^\top {G’}^{-1} \frac{\partial \log \rho_g}{\partial x’}$ holds, so the estimation error is coordinate-invariant.

---

> ### Author Response · Authors · 2022-11-18
> **Reply to reviewer's comments #3**
>
> ### About weaknesses
> > **[W6]** The experiments generally focus on very low-dimensional manifolds (e.g. 3x3 symmetric positive definite matrices). I would expect that autoencoder-style models have most value on high-dimensional data. I suppose that the method has difficulty scaling to high-dimensional data; what is the main bottleneck?
>
> **[A6]**
> - Training our geometrically regularized autoencoders can become computationally more intensive than the vector space regularized autoencoders since we need to calculate various geometric components, such as the geodesic distance, exponential map, and geometric contractive terms. This tendency will not get any better when the data is high-dimensional.
> - To verify how training our geometrically regularized autoencoders scales with the data dimensionality, we have experimentally measured the computational time for training the models using $\text{S}^n$ and $\text{P}(n)$ data with various choices of $n$ in **Appendix D.1.5** and **D.2.6** of the revision, respectively.
> - For the case of $\text{S}^n$, from **Table 6** in **Appendix D.1.5** of the revision, we can observe that applying our geometric components can be scaled to high-dimensional $\text{S}^n$ data reasonably well, except for the geometric contractive term, which requires the derivative of the Jacobian $\frac{\partial \text{r}}{\partial \text{x}} \in \mathbb{R}^{n+1 \times n+1}$ during the update.
> - For the case of $\text{P}(n)$, as $n$ or the data dimensionality $d = n(n+1)/2$ increases, what becomes the main bottleneck is the matrix multiplication and the eigenvalue decomposition required in computing the exponential map, geodesic distance, and geometric contractive term.
> - From **Table 7** in **Appendix D.2.6** of the revision, we can observe that applying our geometric components can be scaled to high-dimensional $\text{P}(n)$ data at a rate slower than the linear rate to $d$, except for the geometric contractive term of which the computational time increases at a nearly quadratic rate to $d$ when applied.
> - To apply our autoencoder regularization method to very high-dimensional non-Euclidean data, we must deal with the geometric components more efficiently with a suitable approximation, even at the cost of some accuracy. For example,  in the experiments in **Section 4.5**, where we consider point cloud data of dimension $nD = 2,048*3 = 6,144$, we have successfully performed the representation learning using GDAE by applying the diagonal approximation to the Riemannian metric as explained in **Appendix D.4.2** of the revision.

---

> ### Author Response · Authors · 2022-12-06
> **Reply to reviewer's comments #4**
>
> ### About minor issues
> > **[M1]** The paper generally talks of Riemannian latent spaces, but, as far as I could tell, mostly Euclidean latent spaces are considered. Perhaps one should here call on one of the many papers which argue that autoencoder-style latent spaces should be viewed as Riemannian (through the pull-back metric). See e.g. https://arxiv.org/abs/1710.11379
>
> **[A-M1]**
> - Following the reviewer's suggestion, we have included some discussions related to the Riemannian latent spaces in **Appendix B.2** of the revision.
> - We have introduced there some recent works on autoencoders that deal with non-Euclidean latent spaces (e.g., hyperspheres [1, 2], hyperbolic spaces or Poincaré balls [3], their mixtures [4], or submanifolds embedded in Euclidean spaces [5]) to better capture the structure of data distributions.
> - We have also provided the formulation of geometric contractive autoencoder (GCAE) in Eq (15) of the revision, a generalization of CAE for non-Euclidean data and latent spaces.
>
> - As the reviewer mentioned, we can consider the pullback of the data space metric via decoder mapping as our latent space metric, according to [6]. It has been observed that applying this metric can better characterize the data distances in latent space and provide more meaningful results in analyzing latent representations.
> - Adopting this metric on the latent space reveals an interesting connection between the GCAE and GRCAE. This pullback metric is obtained as $H(y) = \left(\frac{\partial g}{\partial y}\right)^\top G(g(y)) \left(\frac{\partial g}{\partial y}\right) \in \mathbb{R}^{n\times n}$, and we can observe that the objective functions for GCAE and GRCAE (provided in (15) and (5) of the revision, respectively) become identical when substituting this metric into the GCAE objective. We have summarized this discussion at the end of **Appendix B.2**.
> - References:
> > [1] Davidson, T. R., Falorsi, L., De Cao, N., Kipf, T., & Tomczak, J. M. (2018). Hyperspherical variational auto-encoders. arXiv preprint arXiv:1804.00891.\
> [2] Xu, J., & Durrett, G. (2018). Spherical latent spaces for stable variational autoencoders. arXiv preprint arXiv:1808.10805.\
> [3] Mathieu, E., Le Lan, C., Maddison, C. J., Tomioka, R., & Teh, Y. W. (2019). Continuous hierarchical representations with poincaré variational auto-encoders. Advances in neural information processing systems, 32.\
> [4] Skopek, O., Ganea, O. E., & Bécigneul, G. (2019). Mixed-curvature variational autoencoders. arXiv preprint arXiv:1911.08411.\
> [5] Rey, L. A. P., Menkovski, V., & Portegies, J. W. (2019). Diffusion variational autoencoders. arXiv preprint arXiv:1901.08991.\
> [6] Arvanitidis, G., Hansen, L. K., & Hauberg, S. (2018, February). Latent Space Oddity: on the Curvature of Deep Generative Models. In International Conference on Learning Representations.
>
>
>
> ### About clarity, quality, novelty and reproducibility
> > **[C1]** I found the motivation of the paper to be quite clear, but I struggled with the more mathematical parts of the paper. I would have valued a bit more hand-holding along the way.
>
> > **[Q1]** The work appears sensible, and I particularly enjoyed the denoising autoencoder extension to manifolds. That's very clean and easy to understand. I have a hard time judging the remaining parts of the paper as I didn't quite understand the math.
>
> **[A-C1, Q1]**
> - To address the reviewer's concern about clarity, we have provided some mathematical backgrounds required to formulate our geometric autoencoder regularizations in **Appendix A** of the revision. We hope introducing these backgrounds can help the reviewer and other readers better understand the contents of the paper.

---

> > ### Comment · Reviewer_F6Np · 2022-12-06
> > **Follow-up**
> >
> > Thanks for the detailed feedback. I found the mathematical explanations most helpful, and the revision is clearer. This removes the bulk of my concerns. The key one remaining is that the method seems to have difficulty scaling to high-dimensional manifolds. This could be a matter of implementation but is most likely due to the general computational challenges associated with geometric computations. I appreciate the added efforts to better document this limitation, but at the same time I consider the limitation to be severe as the main use of autoencoders is in dimensionality reduction. I'm happy to increase my score to a 6 due to the improved clarity of the paper, but my positivity is limited by the scalability issue.

---

> > > ### Author Response · Authors · 2022-12-07
> > > **Thank you for the additional comments**
> > >
> > > We appreciate the reviewer's constructive comments. We are glad that the reviewer finds our revision clearer. We agree with the reviewer that scaling to high-dimensional manifolds is of great importance, and addressing the relevant issues will be an interesting topic for future research. We also appreciate the reviewer raising the score of our paper.

---

### Official Review · Reviewer_Ts5i · 2022-10-25

**Confidence:** 3
**Correctness:** 3
**Technical Novelty And Significance:** 3
**Empirical Novelty And Significance:** 3
**Recommendation:** 6

**Clarity, Quality, Novelty And Reproducibility:**

The paper is well-written and very easy to read. The idea of adapting the regularization to non-Euclidean data is novel and interesting.

**Strength And Weaknesses:**

**Strength**
1. The motivation of the paper is clear. It is also very well-organized. I especially like the different notations for points on the manifold and their coordinates.
2. Theorem 1 suggests that the geometric score can be estimated from the trained auto-encoder (a generalization of the result by Alain & Bengio, 2014 in the Euclidean setting), and the numerical results are convincing.

**Weakness**
1. My biggest concern is the increased computational cost after taking into account the geometric information (exponential maps, geodesics, etc.) This seems to be buried under the rug of the paper. I would like to see come comparison of the computational time in training the vanilla autoencoder and the proposed models.
2. The author seems to be using one single chart for a manifold. However, general manifolds are typically not parametrizable using one chart. What would happen if $r(x)$ and $x$ appear in two different charts?
3. It might be a good idea to elaborate, in the main text, what "$\approx$" means in equation (7).
4. Also, is RCAE actively used in the community? There does not seem to be any reference on that in the paper.

**Summary Of The Paper:**

The authors propose geometrically regularized autoencoders for non-Euclidean data. The main contribution of the paper is adapting the regularization terms in the denoising autoencoder (DAE) and the reconstruction contractive autoencoder (RCAE) to the non-Euclidean setting. If the manifold from which the data are sampled are known a priori, the proposed autoencoders achieve improved reconstruction performance, and are able to estimate the score of the non-Euclidean data.

**Summary Of The Review:**

I think this is an interesting paper. I have some concern on the practical implementation and its implication on the computational cost. I am willing to adjust my rating based on the authors' response.

---

> ### Author Response · Authors · 2022-11-18
> **Reply to reviewer's comments #1**
>
> ### About weaknesses
> > **[W1]** My biggest concern is the increased computational cost after taking into account the geometric information (exponential maps, geodesics, etc.) This seems to be buried under the rug of the paper. I would like to see come comparison of the computational time in training the vanilla autoencoder and the proposed models.
>
> **[A1]**
> - We have experimentally compared the computational costs for training the vanilla autoencoders and the models applying our geometric regularization components in **Appendix D.1.5** and **D.2.6** of the revision.
> Specifically, we have measured the computational time spent for a gradient update iteration for the vanilla autoencoder (AE), GDAE, GRCAE, and autoencoders that apply each geometric component (e.g., the geodesic distance, exponential map, and geometric contractive term) to the autoencoder reflecting the non-Euclidean input and output structure (GAE).
> - For the case of $\text{S}^n$, from **Table 6** in **Appendix D.1.5** of the revision, we can observe that the computational time of GDAE and autoencoders applying the geometric components other than the geometric contractive term is not very different from that of the vanilla autoencoder. (We can reasonably scale them to high-dimensional $\text{S}^n$ data.)
> - In the case of $\text{P}(n)$ data, the computational time for geometric components is somewhat longer than vanilla autoencoder due to the frequently required matrix multiplications or eigenvalue decompositions.
> From **Table 7** in **Appendix D.2.6** of the revision, we can observe that the computational time increases at a rate slower than the linear rate to the data dimension $d = n(n+1)/2$ for GDAE and autoencoders applying the geometric components other than the geometric contractive term.
> - For both $\text{S}^n$ and $\text{P}(n)$ data, the GRCAE (which includes the geometric contractive term) require much longer computational time than GDAE since they use the derivative of the reconstruction function Jacobian during training.
>
> > **[W2]** The author seems to be using one single chart for a manifold. However, general manifolds are typically not parametrizable using one chart. What would happen if $r(x)$ and $x$ appear in two different charts?
>
> **[A2]**
> - As explained in **Appendix D.5** of the revision, when we require multiple charts $\\{ (U_1, x_1), \ldots, (U_C, x_C) \\}$ to represent data on a manifold, we can implement and train multiple reconstruction functions defined on each coordinate chart. The objective function for the reconstruction function $r_i: \mathbb{R}^m \rightarrow \mathbb{R}^m$ (this is $\text r_i: U_i \rightarrow \mathcal{M}$ represented in local coordinates) on the $i$-th chart can be defined as in Eq (60) of the revision by applying a weight function $f_i: \mathcal{M} \rightarrow \mathbb{R}_{\geq 0}$ constructed appropriately, e.g., based on partitions of unity. (Please refer to the second paragraph of **Appendix D.5** of the revision for the details.) The final reconstruction results for input $\text{x} \in \mathcal{M}$ can be obtained by geometrically averaging (after appropriate coordinate transformations) the outputs $r_i(x)$ from each mapping with corresponding weights $f_i(x)$.
> - As the reviewer mentioned, for some $\text{x} \in U_i$, $\text{r}_i(\text{x})$ and $\text{x}$ may not be in the same coordinate chart, i.e., $\text{r}_i(\text{x})$ may be outside of $U_i$. In fact, there is no guarantee that $\text{r}_i(U_i) \subseteq U_i$ (or $r_i(x_i(U_i)) \subseteq x_i(U_i)$) would always hold, and the calculation of the objective function (Eq (60) of the revision) or the weighted average of the reconstructed outputs may not be mathematically well-defined when using a single local coordinate.
> - For small $\sigma^2$, such a phenomenon  (if it happens) would mostly occur at $\text{x} \in U_i$ near the boundary of $U_i$ since $r_i$ is trained so that $r_i(x)$ is close enough to $x$, i.e., $r_i(x) - x = \mathcal{O}(\sigma^2)$, according to **Theorem 1** in the manuscript.
> We can eliminate any undesirable effect that this phenomenon may have on the optimization of the objective function in Eq (60) or on the weighted average of the final results by choosing a weight function $f_i$ that has the value of zero in a sufficiently wide region within $U_i$ that includes the boundary of $U_i$.
> - As a side effect, this choice would also assign relatively large weights for the inner regions of $U_i$ far from the boundary of $U_i$ hence inducing a larger regularization effect (e.g., denoising or contraction) in those regions.
> Therefore, this may lead to learning the reconstruction function to direct toward the inner regions of $U_i$, which helps to prevent the range of $\text{r}_i$ (or $r_i$) from going outside of $U_i$ (or $x_i(U_i)$).
> - In the revision, we have summarized the above discussion in the last paragraph of **Appendix D.5** and have indicated at the end of **Section 3** that we discuss these aspects in **Appendix D.5**.

---

> > ### Comment · Reviewer_Ts5i · 2022-12-06
> > **Thank you for your response**
> >
> > I appreciate the authors' detailed response. My concerns have been addressed, and I raise my rating to marginally above accpetence.

---

> > > ### Author Response · Authors · 2022-12-06
> > > **Thank you**
> > >
> > > We appreciate the reviewer providing helpful feedback and raising the rating of our paper.

---

> ### Author Response · Authors · 2022-11-18
> **Reply to reviewer's comments #2**
>
> ### About weaknesses
> > **[W3]** It might be a good idea to elaborate, in the main text, what "$\approx$" means in equation (7).
>
> **[A3]**
> - The approximation means that the two sides differ by an order of $\mathcal{O}(\sigma^2)$. We have clarified this point in the revision.
>
> > **[W4]** Also, is RCAE actively used in the community? There does not seem to be any reference on that in the paper.
>
> **[A4]**
> - To the best of our knowledge, the RCAE, which first appeared in Alain & Bengio (2014), has not been considered much from the practical perspective compared to DAE. We surmise this is because, as also observed in our experiments, the performances of RCAE and DAE are not much different, and DAE is much easier to train. (Please refer to **[A1]** in [Reply to reviewer's comments #1](https://openreview.net/forum?id=_q7A0m3vXH0&noteId=G69o0qkw3DD) where we compare the computational costs of GDAE and GRCAE, of which the tendency is similar to DAE and RCAE.)
> - Also note that when we approximate the objective function based on a randomization method so that RCAE does not require the derivative of Jacobian $\frac{\partial r}{\partial x}$ during training, the formulation results in a form similar to DAE.
> - That is, $E_{\epsilon}[\||r(x+\epsilon) - x\||^2] \approx E_{\epsilon}[\||r(x) + \frac{\partial r}{\partial x} \epsilon - x\||^2] = \||r(x) - x\||^2 + \sigma^2 \text{Tr}\left( \left( \frac{\partial r}{\partial x}\right)^\top \left( \frac{\partial r}{\partial x}\right) \right)$, where $\epsilon \sim N(0, \sigma^2 I)$ and we have used the first-order Taylor's expansion for the first approximation.

---

### Author Response · Authors · 2022-11-18
**Revision uploaded**

We appreciate all the reviewers' constructive comments.
In the revision, we have tried to address each of the criticisms and suggestions offered by the reviewers.
We have highlighted in blue the modifications made from the original manuscript.

### Summary of the major changes in the revision:
- We have refined the example in the introduction to compare the coordinate dependence of vector space autoencoder regularizations and our geometrically regularized autoencoders and have included the experimental details in **Appendix E**.
  - Related comments/answers:
> **[W2]/[A2]** in [Reply to reviewer F6Np's comments #1](https://openreview.net/forum?id=_q7A0m3vXH0&noteId=nOAEwH9Qyr)
- We have provided some mathematical backgrounds on differentiable manifolds and Riemannian geometry required in formulating our geometric autoencoder regularization components in **Appendix A**.
  - Related comments/answers:
> **[W1]/[A1]** in [Reply to reviewer F6Np's comments #1](https://openreview.net/forum?id=_q7A0m3vXH0&noteId=nOAEwH9Qyr)\
> **[C1], [Q1]/[A-C1,Q1]** in [Reply to reviewer F6Np's comments #4](https://openreview.net/forum?id=_q7A0m3vXH0&noteId=hcuhvcd8Ip)\
> **[W1]/[A1]** in [Reply to reviewer hFuw's comments #1](https://openreview.net/forum?id=_q7A0m3vXH0&noteId=9tn1y95Wgsr)
- We have empirically measured the computational time spent for a gradient update step in training our geometrically regularized autoencoders for $\text{S}^n$ and $\text{P}(n)$ data in **Appendix D.1.5** and **D.2.6**, respectively. We have varied the data dimension in the experiments to check how the computational time scales to the dimension.
  - Related comments/answers:
> **[W1]/[A1]** in [Reply to reviewer Ts5i's comments #1](https://openreview.net/forum?id=_q7A0m3vXH0&noteId=G69o0qkw3DD)\
> **[W6]/[A6]** in [Reply to reviewer F6Np's comments #3](https://openreview.net/forum?id=_q7A0m3vXH0&noteId=cns2KNfyZep)

### Summary of the minor changes in the revision:
- In **Appendix B.2**, we have discussed some of the non-Euclidean latent spaces considered in the literature.
  - Related comments/answers:
> **[M1]/[A-M1]** in [Reply to reviewer F6Np's comments #4](https://openreview.net/forum?id=_q7A0m3vXH0&noteId=hcuhvcd8Ip)
- We have refined our ideas dealing with the case of manifolds that require multiple coordinate charts in **Appendix D.5**. We have primarily included a discussion on how to handle the case when $\text{r(x)}$ and $\text{x}$ are not represented in the same coordinate chart.
  - Related comments/answers:
> **[W2]/[A2]** in [Reply to reviewer Ts5i's comments #1](https://openreview.net/forum?id=_q7A0m3vXH0&noteId=G69o0qkw3DD)
- We have discussed the case when the data space Riemannian metric is not known *a priori* in **Appendix D.6**.
  - Related comments/answers:
> **[W3]/[A3]** in [Reply to reviewer hFuw's comments #1](https://openreview.net/forum?id=_q7A0m3vXH0&noteId=9tn1y95Wgsr)
- For other minor changes, please refer to our responses to the reviewers' comments.

---

> ### Author Response · Authors · 2022-12-06
> **Additional discussion**
>
> We would appreciate it if the reviewers kindly check both our revision and replies to the reviewers' comments and provide additional comments if needed.

---

### Decision · Program_Chairs · 2023-01-20

**Decision:**

Accept: poster

**Justification For Why Not Higher Score:**

The contribution is nice but a somewhat limited (generalizing the reconstruction contractive auto encoder to manifolds by considering the generalized Dirichlet energy on manifolds). Does not justify any special highlight at the conference IMO.

**Justification For Why Not Lower Score:**

The reviewers unanimously agreed this paper is above the bar.

**Metareview: Summary, Strengths And Weaknesses:**

This paper initially received two negative (5) and one positive (6) scores.
The main concerns were mostly exposition issues, and missing discussion of computational costs.
These were addressed by the authors and during the rebuttal and discussion period the reviewers with initially negative scores increased their scores to positive (6).
We encourage the authors to discuss the scalability of their method in the final version.

**Note From Pc:**

if the above contains the word "oral" or "spotlight" please see: "oral" presentation means -> notable-top-5% and "spotlight" means -> notable-top-25%. As stated in our emails, we are disassociating presentation type from AC recommendations